# Patient-derived models recapitulate heterogeneity of molecular signatures and drug response in pediatric high-grade glioma

Chen He[1,14], Ke Xu[2,3,14], Xiaoyan Zhu[1,14], Paige S. Dunphy[1,4,14], Brian Gudenas[1], Wenwei Lin[5], Nathaniel Twarog[5], Laura D. Hover[1], Chang-Hyuk Kwon[1], Lawryn H. Kasper[1], Junyuan Zhang[1], Xiaoyu Li[6], James Dalton[6], Barbara Jonchere[7], Kimberly S. Mercer[7], Duane G. Currier[5], William Caufield [8], Yingzhe Wang [8], Jia Xie[5], Alberto Broniscer[9], Cynthia Wetmore[10], Santhosh A. Upadhyaya[4], Ibrahim Qaddoumi[4], Paul Klimo[11], Frederick Boop[11], Amar Gajjar[4], Jinghui Zhang [3], Brent A. Orr [6], Giles W. Robinson [4], Michelle Monje [12], Burgess B. Freeman III[8], Martine F. Roussel [7], Paul A. Northcott [1], Taosheng Chen [5], Zoran Rankovic[5], Gang Wu [2,3], Jason Chiang [6✉], Christopher L. Tinkle [13✉], Anang A. Shelat [5✉] & Suzanne J. Baker [1✉]

Pediatric high-grade glioma (pHGG) is a major contributor to cancer-related death in children. In vitro and in vivo disease models reflecting the intimate connection between developmental context and pathogenesis of pHGG are essential to advance understanding and identify therapeutic vulnerabilities. Here we report establishment of 21 patient-derived pHGG orthotopic xenograft (PDOX) models and eight matched cell lines from diverse groups of pHGG. These models recapitulate histopathology, DNA methylation signatures, mutations and gene expression patterns of the patient tumors from which they were derived, and include rare subgroups not well-represented by existing models. We deploy 16 new and existing cell lines for high-throughput screening (HTS). In vitro HTS results predict variable in vivo response to PI3K/mTOR and MEK pathway inhibitors. These unique new models and an online interactive data portal for exploration of associated detailed molecular characterization and HTS chemical sensitivity data provide a rich resource for pediatric brain tumor research.

[1] Department of Developmental Neurobiology, Memphis, TN, USA. [2] Center for Applied Bioinformatics, Memphis, TN, USA. [3] Department of Computational Biology, Memphis, TN, USA. [4] Department of Oncology, Memphis, TN, USA. [5] Department of Chemical Biology and Therapeutics, Memphis, TN, USA. [6] Department of Pathology, Memphis, TN, USA. [7] Department of Tumor Cell Biology, Memphis, TN, USA. [8] Preclinical Pharmacokinetics Shared Resource St. Jude Children's Research Hospital, Memphis, TN, USA. [9] Division of Hematology-Oncology, Children's Hospital of Pittsburgh, Pittsburgh, PA, USA. [10] Exelixis, Inc., Alameda, CA, USA. [11] Department of Surgery, St. Jude Children's Research Hospital, Memphis, TN, USA. [12] Department of Neurology, Stanford University, Stanford, CA, USA. [13] Department of Radiation Oncology, St. Jude Children's Research Hospital, Memphis, TN, USA. [14]These authors contributed equally: Chen He, Ke Xu, Xiaoyan Zhu, Paige S. Dunphy. ✉email: Suzanne.Baker@StJude.org; Anang.Shelat@StJude.org; Christopher.Tinkle@StJude.org; Jason.Chiang@StJude.org

Brain tumors are the predominant cause of cancer-related morbidity and mortality in children[1]. Pediatric diffuse high-grade gliomas (pHGG) comprise approximately 20% of all childhood brain tumors. This heterogeneous group of tumors carries a devastating prognosis, with 70–90% of patients dying within 2 years of their diagnosis[1]. Genome-wide analyses have transformed our understanding of pHGGs to illuminate distinct molecular features compared to adult HGG, including a close association between tumor location, patient age, and recurrent mutations that indicates an intimate connection between pHGG pathogenesis and developmental context[2,3]. For example, histone H3K27M mutations, which are rare in other tumor types, occur in approximately 80% of diffuse intrinsic pontine gliomas (DIPG) and other diffuse HGGs in midline structures such as the thalamus[4–7]. This striking association has redefined the diagnosis of these tumors, with the 2016 World Health Organization Classification of Tumors of the Central Nervous System (CNS) now incorporating molecular-based criteria to define diffuse midline glioma—H3K27M mutant (DMG-K27M) as a distinct diagnostic entity[8]. H3K27M mutations most frequently occur in 2 of the 15 genes that encode histone H3, with *H3F3A* encoding H3.3 K27M and *HIST1H3B* encoding H3.1 K27M in approximately 75% and 25% of H3 mutant DIPG, respectively[2]. Activating mutations in the gene encoding the BMP receptor ACVR1 are found almost exclusively in DIPG and preferentially co-occur with H3.1 K27M mutations, generally in younger patients, demonstrating an even more restricted association with developmental context[5,9–12]. In contrast, histone H3.3 G34R/V mutations are found in approximately 15% of cerebral cortical HGG, with patient age ranging from older adolescents through young adulthood[5–7].

Variable combinations of additional mutations also contribute to intertumoral heterogeneity, including mutations activating the receptor tyrosine kinase–RAS–PI3-kinase pathway, alterations inactivating tumor suppressors *TP53* or *CDKN2A*, mutations in epigenetic regulators such as ATRX, and others[5,6,9–12]. Different subclasses of pHGG are readily detected through comparisons of genome-wide DNA methylation profiles, which may reflect both the developmental origins of the tumors and the consequences of tumorigenic mutations[13]. This epigenetic characterization allows a refined molecular classification of CNS tumors and is increasingly being incorporated into clinical practice.

Despite rapid advances in the characterization of the genomic and epigenetic landscape of pHGG, effective therapeutic approaches are still lacking for almost all pHGG patients[14]. In vitro and in vivo models that recapitulate the complexity and heterogeneity of pHGG are essential to advance our understanding and identify therapeutic vulnerabilities of these deadly childhood brain tumors.

Here, we report the establishment of a unique collection of 21 patient-derived orthotopic xenograft (PDOX) models and 8 new pHGG cell lines recapitulating molecular signatures of the primary tumors from which they were derived and representing a broad spectrum of the heterogeneity found in pHGG. We use a total of 14 pHGG cell lines and two control cell lines for high-throughput screening (HTS) to identify drug sensitivity and validate the in vitro heterogeneity of response for two drugs in vivo. Detailed molecular characterization of these novel models and the results from HTS chemical sensitivity studies on the large cell line panel are available through an interactive online data portal providing a rich resource to the pediatric brain tumor research community.

## Results

### Patient-derived orthotopic xenografts and cell lines of pHGG represent diverse tumor subtypes. We generated a unique

resource of 21 pHGG PDOX by implanting dissociated tumor cells from surgical or autopsy samples into the brains of immunodeficient mice. Patient age at diagnosis ranged from 4 to 19 years, with a median age of 12 years and a median survival of 12 months (Supplementary Data 1). Engraftment efficiency was higher for surgical samples (56%) than for autopsy samples (33%), likely due to decreased tumor cell viability in post-mortem material. After initial tumor engraftment, tumors were collected, dissociated, and passaged by intracranial implantation into additional immunodeficient mouse hosts to confirm that the PDOX could be reliably maintained, and for expansion and banking. Passaged tumors were dissected and cryopreserved as viable cells, snap-frozen for subsequent molecular analyses, or processed for histopathology evaluation. The majority of PDOX models were transduced with lentivirus expressing luciferase-2a-YFP to enable in vivo imaging. The mouse host survival times for this collection of PDOX models ranged from 2 to 6 months after intracranial implantation (Supplementary Data 1). For a subset of the tumors, dissociated cells were adapted for in vitro propagation under neural stem cell conditions to facilitate the use of these as matched cell line models for high-throughput drug screening (Fig. 1).

We performed DNA methylome profiling to classify all 21 PDOX tumors, 19 matched patient tumors from which PDOXs were established, and 8 matched cell lines established from PDOXs (Supplementary Data 1b). Using t-distributed stochastic neighbor embedding (t-SNE) analysis of methylation profiles from these samples and a reference set including samples from diverse brain tumor subgroups[13], all PDOX models and cell lines clearly grouped with glioblastoma or glioma, distinct from embryonal or ependymal tumors. (Fig. 2a). An expanded view of the glioblastoma and glioma clusters shows that 17 of 19 PDOXs and all 8 cell lines cluster closely with the tumor from which they were derived (Fig. 2b). Following the established classification scheme[13], this novel collection of PDOX models comprises 6 DMG-K27M, 2 pleomorphic xanthoastrocytomas (PXA), 10 glioblastoma, IDH wild type, including three H3.3 G34 mutant (GBM G34), four subclass midline (GBM Mid), one subclass RTK II (pedRTKII), and two subclass RTKIII (ped-RTKIII). Three tumors did not directly match the reference clusters, but were closest to GBM Mid, pedRTKII, or PXA (Fig. 2 and Supplementary Data 1a, b). For two models, SJ-HGGX51 and SJ-HGGX78, the diagnostic patient tumor from which the PDOX was established clustered with reference samples from control inflammatory cells (CONTR_INFLAM), suggesting the presence of large areas of necrosis in the tumor surgical specimens with exuberant inflammatory infiltration, while the PDOX clustered with, or very close to the pedRTKII glioblastoma subgroup. Three PDOX models, SJ-HGGX56, SJ-HGGX58, and SJ-HGGX59 that were established from patient tumors with mismatch repair deficiency (MMRD), clustered tightly with their matched patient tumors, and close to one another, along with midline tumors despite their cortical location in the patients (Fig. 2b). In addition to the eight cell lines established from these PDOX models, we also evaluated methylation profiles from six previously reported DIPG cell lines[15] used in our preclinical testing experiments. All H3K27M mutant tumors, including patient tumors, PDOXs, and cell lines, clustered with the DMG-K27M subgroup as expected (Fig. 2a, b).

Histopathology of PDOX models showed typical characteristics of pediatric HGG, including high cellularity, varying extent of astrocytic differentiation, readily apparent mitotic activity, infiltration of the CNS parenchyma, vascular endothelial proliferation and areas of necrosis. PDOX models recapitulated architectural and cytologic features seen in the corresponding patient tumors from which they were derived (Fig. 3).

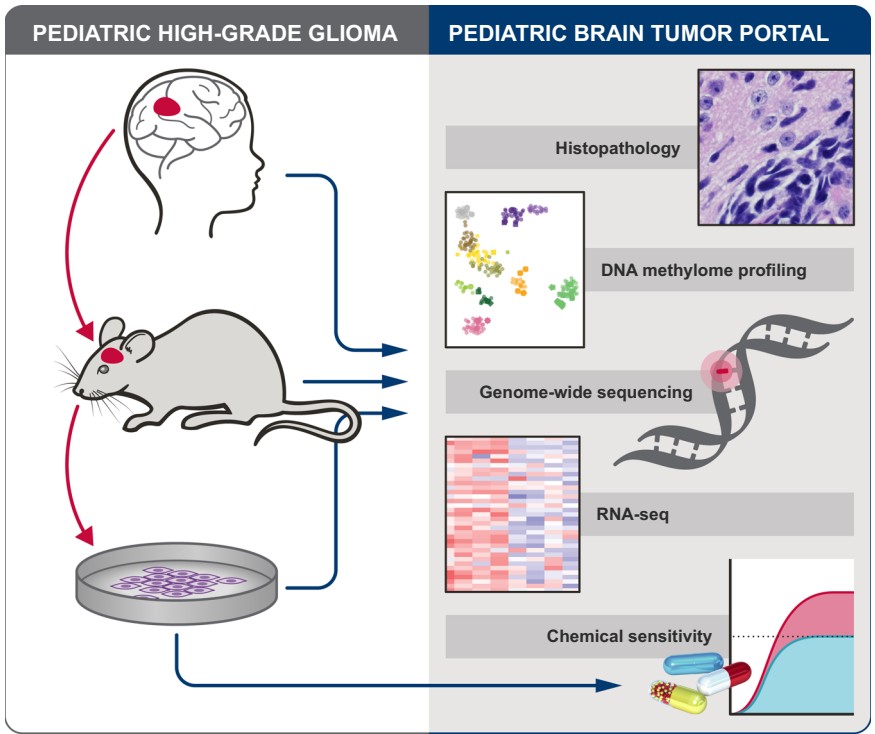

**Fig. 1 Overview of PDOX and cell line establishment, characterization and preclinical testing, and pHGG data available in Pediatric Brain Tumor Portal.** Patient high-grade glioma samples were directly implanted into recipient mouse brain and passaged as PDOX models. Cell lines were also established from a subset of PDOX models (left). Each PDOX or cell line, and when available, matched patient tumor, were evaluated for histopathology, DNA methylome profiles, genome-wide sequencing and RNAseq. Genome sequencing of matched normal reference is also available for most PDOX models. Chemical sensitivity testing was performed with cell lines (right). All associated data is available through an interactive data portal, the Pediatric Brain Tumor Portal (pbtp.stjude.cloud).

**PDOX and cell lines recapitulate recurrent mutations and gene expression signatures characteristic of pHGG.** For a more comprehensive view of the genomic landscape of these models, we performed whole-genome (WGS) or whole-exome sequencing (WES) on all PDOX, matched patient tumors and derived cell lines. Somatic mutations and potentially pathogenic germline mutations were identified for tumors with matched germline samples for 16/21 lines, and potentially pathogenic non-silent mutations were annotated for tumors without available matched germline (Fig. 4 and Supplementary Data 2). As expected, there was a dramatically increased mutation burden in PDOXs with MMRD (SJ-HGGX56, 58 and 59) compared to the rest of the cohort (median non-silent SNVs in the exome of 19,336 compared to 26). Recurrent mutations characteristic of pHGG were well-represented in this cohort of tumors, including hotspot mutations in histone H3, as well as mutations in genes involved in chromatin and transcription regulation such as *ATRX*, *BCOR* and *MYCN*, recurrent alterations in the receptor tyrosine kinase–RAS–PI3-Kinase pathway, including missense mutation, amplification and gene fusion of *PDGFRA*, alterations in the *TP53* and RB/cell cycle pathways, and activating missense mutation in *ACVR1* (Fig. 4). The great majority of PDOXs and cell lines maintained signature mutations and copy-number abnormalities (CNAs) found in the matched patient tumors including large-scale gains and losses and focal amplifications in extra-chromosomal DNA, although there were some examples of divergence between patient tumor and derivative models (Fig. 4 and Supplementary Figs. 1 and 2). PDOX models also maintained mutations of signature glioma genes over multiple passages, as shown for passages 7 and 10 of SJ-DIPGX7 and passages 3 and 4 of SJ-DIPGX29 (Fig. 4).

We previously showed that pHGG, including DIPG, showed heterogeneous expression signatures recapitulating the glioma subgroups proneural, proliferative, and mesenchymal[16,17]. Analysis of these expression signatures in the entire cohort of patient tumors, PDOXs, and cell lines showed PDOX models represented in all three expression subgroups. However, proliferative signatures were much stronger in general in PDOX and cell line models compared with patient tumors (Fig. 5). Consistent with this observation, analysis of genes that were differentially expressed between PDOX and patient samples across the entire matched cohort ($|\text{logFC}| > 1$, adj. $p < 0.05$) showed upregulation of genes associated with cell cycle progression (adj. $p < 2.2\text{e}{-}16$) and downregulation of genes associated with inflammatory response (adj. $p < 2.2\text{e}{-}16$) in PDOX (Supplementary Data 3). We removed the shared differences between PDOX and patient tumors to better compare the similarity in expression signatures between matched PDOX and patient tumors and found that expression signatures of PDOX correlated strongly with their matched patient tumor (Supplementary Fig. 3). Representative PDOX models retained fidelity of transcriptome signatures over multiple passages (Pearson correlation 0.98, $p < 2.2\text{e}{-}16$). Cell lines, which represent extensive passaging in neural stem cell growth media, also showed strong fidelity with the matched PDOX models from which they were derived (Pearson correlation from 0.87 to 0.95, $p < 2.2\text{e}{-}16$) (Supplementary Fig. 4).

**Online interactive portal of PDOX and cell line characterization.** To maximize the utility of these well-characterized models for the cancer research community, we developed an online Pediatric Brain Tumor Portal (https://pbtp.stjude.cloud) that supports interactive exploration of all molecular data. Within this rich open-access portal, detailed information can be viewed or downloaded for each individual model, summarizing clinical, molecular, and histopathological features (Supplementary Fig. 5).

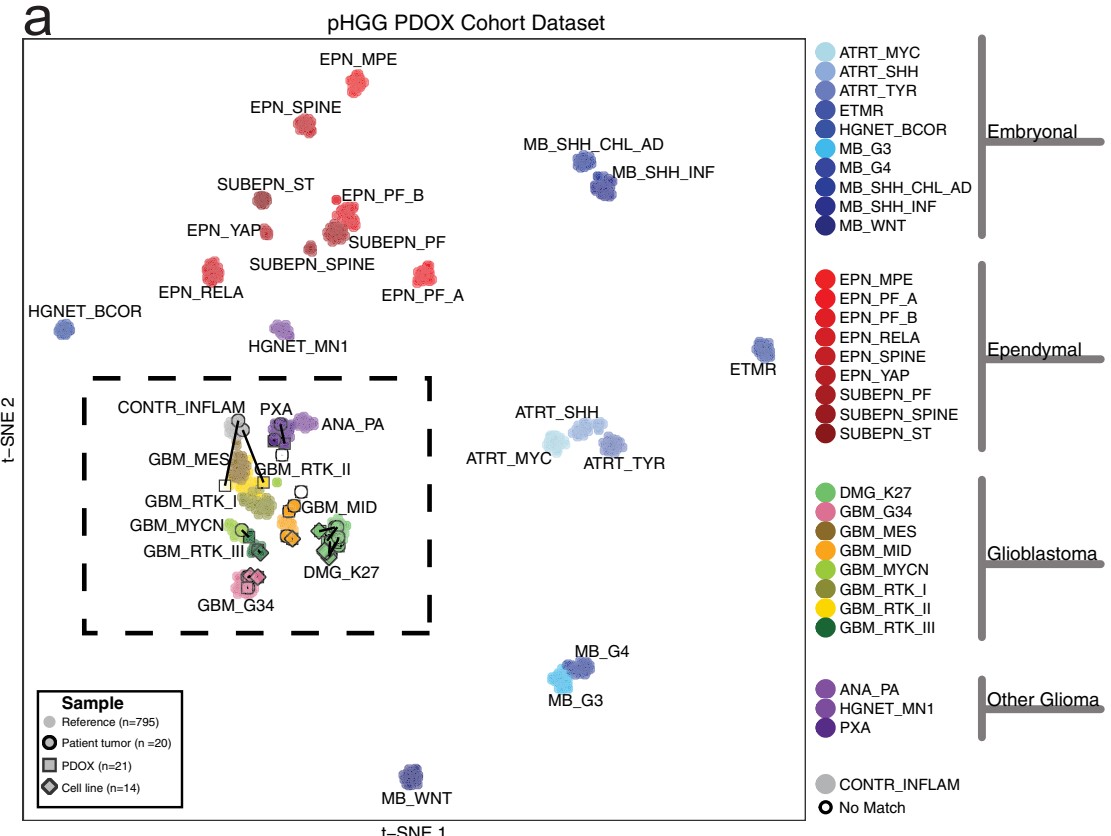

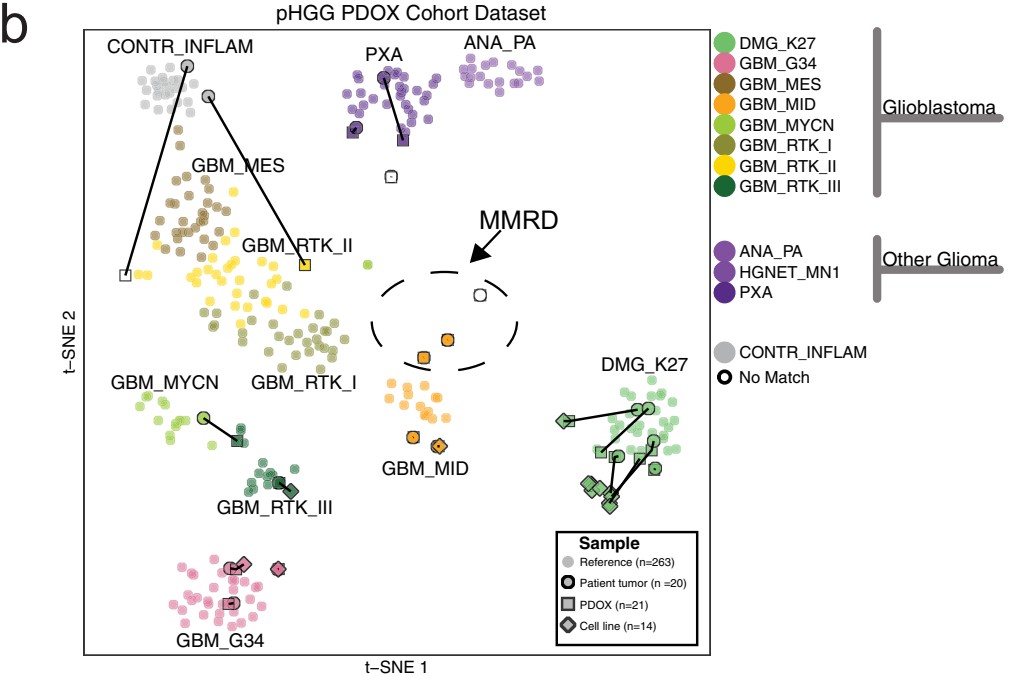

Models can be obtained by completing a request form through the portal. In addition, this portal will house information about other pediatric brain tumor PDOX models derived at St. Jude Children's Research Hospital[18] and will be continually updated with new models in the future.

**High-throughput screen for drug response of different pHGG subtypes.** We selected a panel of 16 cell lines to assess drug sensitivity of different subgroups of pHGG: eight glioma cell lines were established from PDOXs reported here, six were previously reported patient-derived DIPG cell lines[19,20] (Supplementary

**Fig. 2 DNA methylation classification of patient tumors is conserved in corresponding PDOXs and cell lines. a** t-SNE plot showing tumor subgroups based on DNA methylation profiling. Nineteen patient tumors (circles), 21 PDOXs (squares), and 14 cell lines (diamonds) are outlined in black. Lines connect PDOXs and cell lines with the patient tumors from which they were derived. Tumor subgroup classifications are color-coded. Circles without outlines are reference samples from Capper et al. Dashed square shows region containing all HGG samples. Classifications: Embryonal tumors: atypical teratoid rhabdoid tumors (ATRT), embryonal tumor with multilayered rosettes (ETMR), high-grade neuroepithelial tumor with *BCOR* alteration (HGNET_BCOR), and medulloblastoma (MB). Ependymal tumors: ependymoma (EPN), subependymoma (SUBPEN), myxopapillary ependymoma (MPE), posterior fossa (PF), supratentorial (ST). Glioblastoma: diffuse midline glioma (DMG) and glioblastoma (GBM). Other glioma: anaplastic pilocytic astrocytoma (ANA_PA), high-grade neuroepithelial tumor with *MN1* alteration (HGNET_MN1), anaplastic pleomorphic xanthoastrocytoma (PXA). Control tissue, inflammatory tumor microenvironment (CONTR_INFLAM). **b** Enlarged view of boxed region in **a**. Dashed oval shows patient tumors with MMRD and derived PDOX models.

Table 1). Two different sources of proliferating normal human astrocytes were used as controls: astrocytes generated from iPS cells (iAstro) and astrocytes isolated from human embryonic brainstem (HABS) (Supplementary Table 1). Together, the cohort comprised 2 cortical HGG with H3.3 G34R mutation, 2 HGG with wild-type histone H3, and 10 DIPG; 7 with H3.3 K27M and 3 with H3.1 K27M mutation. We optimized all cells to grow in 384-well format plates for HTS and conducted a series of validation experiments to ensure reproducibility, including cell proliferation curves, culture conditions, and the automation parameters of HTS. We showed that DIPG cells plated as adherent cultures on basement membrane matrix compared with the same cells grown as tumorspheres responded similarly to a panel of 53 different drugs representing a range of mechanisms of action (MOA) tested in a dose–response (DR) format (Supplementary Fig. 6, Pearson correlation = 0.994; Supplementary Data 4a, b). Therefore, we conducted HTS experiments with adherent cells to avoid variability within and between cultures introduced by variable tumorsphere sizes.

Top hits from an initial screen of 1134 FDA-approved drugs at a single concentration in nine pHGG cell lines (Supplementary Data 4c) were supplemented with recently FDA-approved oncology drugs, epigenetic modulators, clinical candidates, and relevant chemical probes to yield a set of 93 drugs that were screened against the full panel of 16 cell lines in DR format using an independently prepared compound plate. For all 16 cell lines, the $z'$ values[21] calculated from 310 384-well assay plates showed an excellent signal to noise ratio ($z'$ values: median = 0.82, minimum = 0.56, maximum = 0.92) (Supplementary Fig. 7). An additional 246 compounds, sampling a broader range of drug MOA, were screened in DR format in four exemplar pHGG models representing different histone subtypes. Results from all stages of these HTS studies are presented in Supplementary Data 4 and are available online for exploration in the Pediatric Brain Tumor Portal (pbtp.stjude.cloud) where interactive features allow the user to query by drug class, specific compound name, tumor subgroup, or tumor cell lines, and to visualize results in multiple formats, including data on the range of responses to selected compounds, the sensitivity to all tested drugs for selected cell lines, and customized overlay dose–response curves. A mouseover feature also allows the user to identify outliers and view compound information, dose–response values, and the associated dose–response curve.

Results from the comprehensive screen of 93 compounds in 16 cell lines are summarized in Fig. 6. To highlight the most selectively effective drug for each cell line, we calculated the area under the curve (AUC) from the DR of each drug and subtracted the median AUC for that drug calculated over all cell lines (Fig. 6a). Notably, the distribution of responses for SUDIPG-XXI showed that these cells were more sensitive than other lines to nearly every drug tested. Consequently, we flagged this model as an outlier and removed it from subsequent analyses. Several drugs were identified as the most selective for

more than one pHGG model: the NF-κB inhibitor EVP4593 for SJ-DIPGX7c and SUDIPG-XIII, the HSP90 inhibitor tanespimycin for SJ-DIPGX37c and SJ-HGGX39c, and the MEK inhibitor trametinib for SJ-DIPGX9c and SUDIPG-XIX. The GSK3 inhibitor LY2090314 was the most selective drug for both SUDIPG-IV and HABS control cells, whereas the proteasome inhibitor marizomib was the most selective for SUDIPG-VI and iAstro control cells.

We compared the distribution of AUC values for trametinib, an FDA-approved drug, across all 15 models to AUCs for other drugs in the set that have been evaluated in clinical trials or implicated as promising agents in preclinical studies for adult or pediatric glioma (Fig. 6b, c and Supplementary Fig. 8a). THZ1, which targets CDK7 (ref. [22]), was the most active drug (highest median AUC) in this subset and was more selective for pHGG models compared to iAstro and HABS. The CDK9 inhibitor CDK9-IN-2 (ref. [23]) had lower AUC values compared to THZ1, but also showed selectivity for most pHGG models relative to control cell lines. CDK7 and CDK9 are key regulators of transcription initiation and elongation, respectively, supporting the concept of targeting transcriptional dependencies in tumor cells[24]. In contrast, the pan-BET inhibitor JQ1 (ref. [25,26]), which also impacts transcription regulation, was much weaker than the CDK7 and CDK9 inhibitors, and was equally or more cytotoxic to control cell lines (Fig. 6b). The anti-metabolite GMX-1778, which disrupts the regeneration of NAD$^+$ via NAMPT, showed strong efficacy, in agreement with a previous report[27]. However, we did not observe enhanced sensitivity in a line with *PPM1D* mutation (SJ-DIPGX37c) as predicted by a previous study[27]. Consistent with previous reports that evaluated broad-spectrum HDAC inhibitors[15,19,28,29], panobinostat showed strong efficacy in vitro. However, CUDC-907, a dual-acting inhibitor of class I PI3K and HDAC[30], had higher median AUC and was slightly more selective for pHGG models. The proteasome inhibitor marizomib is being evaluated in a phase I combination study with panobinostat in DIPG (NCT04341311)[15]. This drug had the largest interquartile range of AUC values in the subset and induced significant cytotoxicity in iAstro cells (Fig. 6b and Supplementary Fig. 8a).

Inhibitors of PI3K/mTOR (omipalisib) and MEK (trametinib) were selective for the same three DIPG pHGG models over other cell lines (Fig. 6c). However, trametinib was generally much less active in the panel as evidenced by its significantly lower median AUC (Fig. 6b). A transcriptional MAPK Pathway Activation Score (MPAS) was previously developed based on expression of a select set of genes that correlate with sensitivity to MAPK pathway inhibitors in cell lines from some, but not all tumor types[31]. We determined the MPAS in the 14 cell lines evaluated in our drug screen but found no correlation between MPAS and response to trametinib (Pearson correlation −0.34) (Supplementary Fig. 8b), showing substantial heterogeneity in pHGG response without a simple relationship to gene expression signatures.

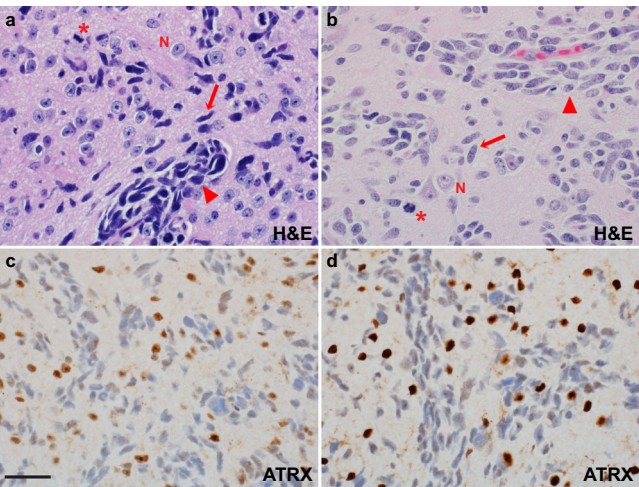

**Fig. 3 Histopathology of PDOX recapitulates salient features of the patient tumor from which it was derived.** H&E staining of HGG PDOX SJ-HGGX6 (**a**), shown as a representative example, recapitulates histologic features of its corresponding primary human tumor (**b**), including infiltration of the CNS parenchyma (arrow), perivascular invasion (black right pointing triangle), and apparent mitotic activity (asterik). Nuclear ATRX immunoreactivity, while retained in the entrapped neurons, is lost in the PDOX tumor cells (**c**), as in its corresponding primary human tumor (**d**). N: entrapped cerebral cortical neurons. Scale bar: 50 μm. H&E staining was performed for 21 PDOX models, 20 with matching patient tumor. ATRX IHC was performed in six PDOX/patient tumor pairs, four with ATRX mutation, and two without ATRX mutation. Loss of ATRX immunoreactivity was consistent with mutation status in all samples tested.

The WEE1 inhibitor MK-1775 (adavosertib), which is being evaluated clinically with radiotherapy in pHGG (NCT01922076), showed low efficacy in our panel. Likewise, the alkylating agent temozolomide (TMZ), a standard of care in adult gliomas, was ineffective in our tumor models, consistent with the lack of clinical response to TMZ in pHGG[32]. We also tested two other alkylating agents, streptozocin and nimustine[33], and both were ineffective in our pHGG models (Supplementary Data 4c, e).

Finally, to better explore patterns of drug sensitivity across the cell lines in our panel, we performed unsupervised clustering of cell lines (columns) and compounds (rows) with all 93 compounds (Supplementary Fig. 8c) or using the top 25% most active drugs out of the 93 evaluated (Fig. 6d). Notably, tumor models did not cluster according to histone subtype based on their responses to the drugs. We observed two compound clusters that were enriched for molecules acting by the same MOA indicating some selective sensitivity in response to inhibition of particular processes. Cluster 1 contained four proteosome inhibitors (marizomib, ONX-0914, ixazomib, and bortezomib) and Cluster 2 contained four compounds targeting transcriptional dependencies (THZ1, CDK9-IN-2, panobinostat and CUDC-907) and two dual activity compounds including PI3K inhibition (CUDC-907 and omipalisib). Cluster 2 was characterized by a significant lack of efficacy against the iAstro control cell line, therefore showing some selectivity for HGG.

**Inhibition of PI3K/mTOR and MEK signaling show selective effects alone and in combination in pHGG in vitro and in vivo.** We sought to determine if sensitivity differences between tumor cell lines detected by HTS translated into in vivo effects in orthotopic brain tumors. For these studies, we chose PI3K/mTOR and MEK pathway inhibitors because responses in different pHGG lines varied and target engagement can be reliably

detected in tumor tissue. HTS results showed that the PI3K/mTOR inhibitor omipalisib was active in most pHGG models but varied in potency. In contrast, the MEK inhibitor trametinib showed weak activity in many tumor cells (Fig. 6 and Supplementary Data 4f). While these pathways are compelling targets for treatment given known genetic aberrations in both pathways in pHGG[2,5], both compounds are known substrates of ATP-dependent efflux pumps P-gp and BCRP, limiting their brain exposure[34,35]. Therefore, we selected paxalisib (GDC0084) and mirdametinib (PD0325901), which target PI3K/mTOR and MEK, respectively, and have superior blood–brain barrier penetration and exposure[34,36,37], to validate the effects of pathway inhibition for in vitro/in vivo comparisons.

Numerous studies have shown extensive cross-talk between these pathways and promise for enhanced efficacy of combined inhibition approaches[38,39]. We conducted in vitro quantitative synergy assays in seven cell lines and analyzed results using the Bivariate Response to Additive Interacting Doses (BRAID) response surface model[40]. BRAID $\kappa$ denotes the type of interaction ($\kappa < 0$ is antagonistic; $\kappa = 0$ is additive, and $\kappa > 0$ is synergistic), whereas BRAID $IAE_{50}$ computes the degree to which a combination achieves a minimal efficacy within a defined concentration range (in this study, 50% reduction in cell viability for concentrations $\leq 1$ μM). Higher $IAE_{50}$ means that the combination is more efficacious. The combination of paxalisib and mirdametinib exerted synergistic growth inhibition ($\kappa > 0$) in three pHGG cell lines (SJ-DIPGX29c, SJ-DIPGX37c, and SJ-HGGX6c) and iAstro controls, but not in SJ-DIPGX7c, SJ-HGGX42c, SJ-HGGX6c, or HA-bs controls (Fig. 7). When considering the efficacy of the combination ($IAE_{50}$), synergy made a major contribution in the case of SJ-DIPGX37c and SJ-DIPGX29c, as indicated by the curvature of the 50% (black) and 90% (white) cell viability isoboles. In contrast, the $IAE_{50}$ value for SJ-DIPGX7c was driven solely by paxalisib: the 50% and 90% isoboles run parallel to the $y$-axis because mirdametinib exerts little cytotoxicity on its own and fails to potentiate the activity of paxalisib. While the synergy is highest in iAstro ($\kappa = 8.8$), as evidenced by the clear shift in the 50% isobole with increasing mirdametinib concentrations, both drugs are weakly cytotoxic on their own and their interaction is insufficient to induce high combined efficacy ($IAE_{50} = 1.0$).

We selected two H3.3 K27M mutant DIPG PDOX models, SJ-DIPGX37 and SJ-DIPGX7, to test whether in vitro HTS results predicted in vivo response. Both harbor mutations targeting PI3K and *TP53*-related pathways (Fig. 4), and their corresponding cell lines (SJ-DIPX7c and SJ-DIPGX37c) demonstrated different in vitro responses. Both cell lines showed similar sensitivity to paxalisib ($EC_{50} = 0.32$ μM for SJ-DIPGX37c vs. 0.22 μM for SJ-DIPGX7c), whereas mirdametinib induced a stronger response ($EC_{50} = 1$ μM) in DIPGX37c, which has *PIK3R1* and *PPM1D* mutations and did little in SJ-DIPGX7c, which has *PIK3CA* and *TP53* mutations (Figs. 4 and 7). Similar responses were observed with trametinib (Supplementary Data 4f).

We evaluated PI3K and MEK pathway inhibition in intracranial PDOX models matched to these cell lines by evaluating levels of p-AKT S473 and p-ERK T202/Y204, respectively (Fig. 8 and Supplementary Fig. 9). Interestingly, while the levels of PI3K pathway activity, as assessed by p-AKT, were similar between both PDOXs, SJ-DIPGX7 showed markedly lower levels of p-ERK, indicating lower MEK pathway activation (Fig. 8a). Consistent with a reduced reliance on MEK signaling, mirdametinib did not significantly alter cell survival or proliferation in SJ-DIPGX7, as assessed by active caspase-3 and phospho-histone H3, respectively. Paxalisib treatment alone induced a trend of increased cell death, and when combined with mirdametinib significantly increased cell death (Fig. 8a).

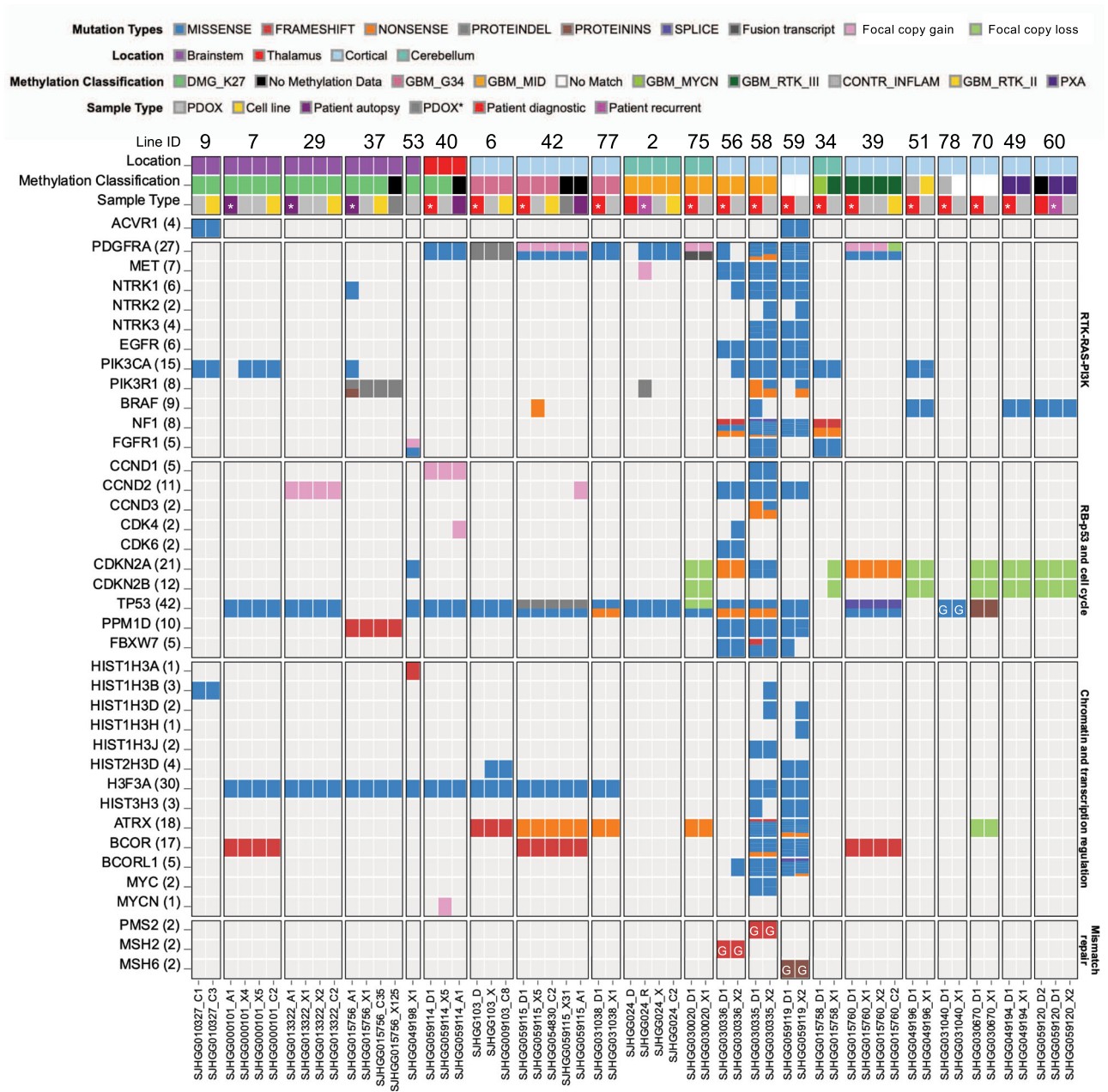

**Fig. 4 Genomic landscapes of PDOX and cell lines conserve alterations present in the matched patient tumor and represent a variety of pHGG subtypes.** Alterations in genes recurrently mutated in pHGG are indicated on the left. Pathways are indicated on the right. Columns show tumor samples. PDOX and cell lines are grouped together with patient tumors from which they were derived. Numbers are the PDOX identifier IDs across the top, sequence file IDs across the bottom. Rows show the location of patient tumors, DNA methylome classification, and tumor sample type. In some cases, patient tumor samples from recurrence or autopsy are included along with the diagnostic sample. Asterisk in tumor sample type indicates the patient tumor from which the PDOX was derived. For 7, passages 7 and 10 of SJ-DIPGX7 are shown, and for 29, passages 3 and 4 of SJ-DIPGX29 are shown. PDOX* (dark gray box) indicates xenograft generated by implanting the associated cell line. Mutations in signature genes shown as rows are indicated by Mutation Type color code. G in block indicates germline mutation.

SJ-DIPGX37 showed a significant decrease in tumor cell proliferation in vivo with paxalisib treatment, and a greater magnitude effect with mirdametinib, while neither induced significant tumor cell death. Strikingly, the combination of paxalisib and mirdametinib significantly enhanced tumor cell death in vivo beyond levels induced by either agent alone (Fig. 8b). Consistent with in vitro synergy studies, the combination had a much more significant impact on cell death in SJ-DIPGX37.

We further extended these promising results in SJ-DIPGX37 to evaluate effects on the survival of tumor-bearing mice. We

reduced doses of paxalisib to 8 mg/kg and mirdametinib to 14 mg/kg in monotherapy controls and in the combination for long-term treatment due to significant weight loss at higher dosages. These doses still effectively inhibited both pathways in the brain, with the combination showing slightly enhanced suppression of p-ERK1/2 relative to mirdametinib alone (Supplementary Fig. 9c). We randomly distributed 24 mice bearing SJ-DIPGX37 PDOX into four arms: (1) vehicle, (2) paxalisib, (3) mirdametinib, or (4) combination and found that the combination of paxalisib and mirdametinib, but not

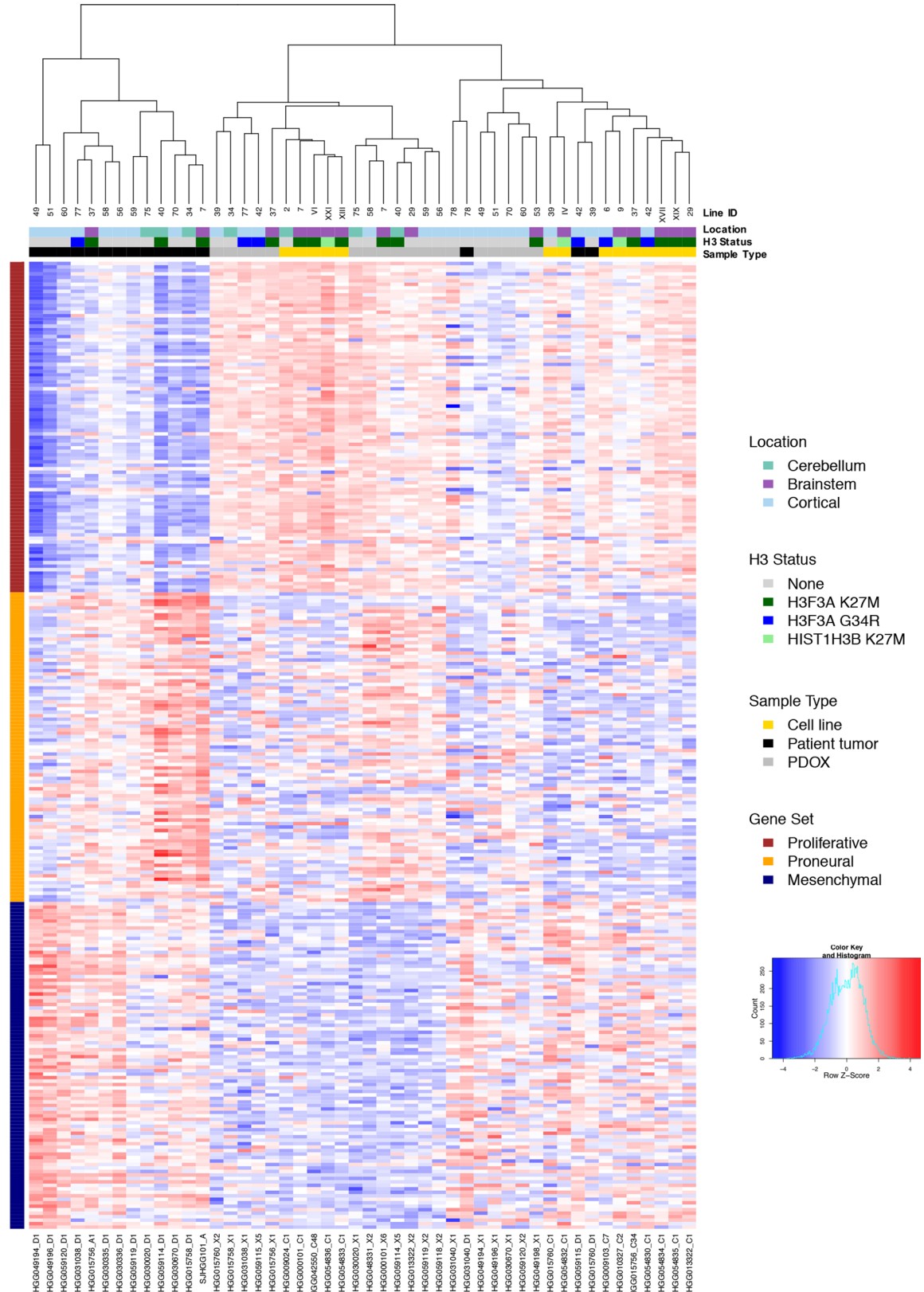

**Fig. 5 Gene expression signatures of PDOX models recapitulate glioma expression subgroups.** Unsupervised hierarchical clustering of RNA-seq quantification (log CPM) of genes from three expression signatures recapitulating glioma subgroups proliferative, proneural, and mesenchymal across the patient tumors, PDOXs, and cell lines.

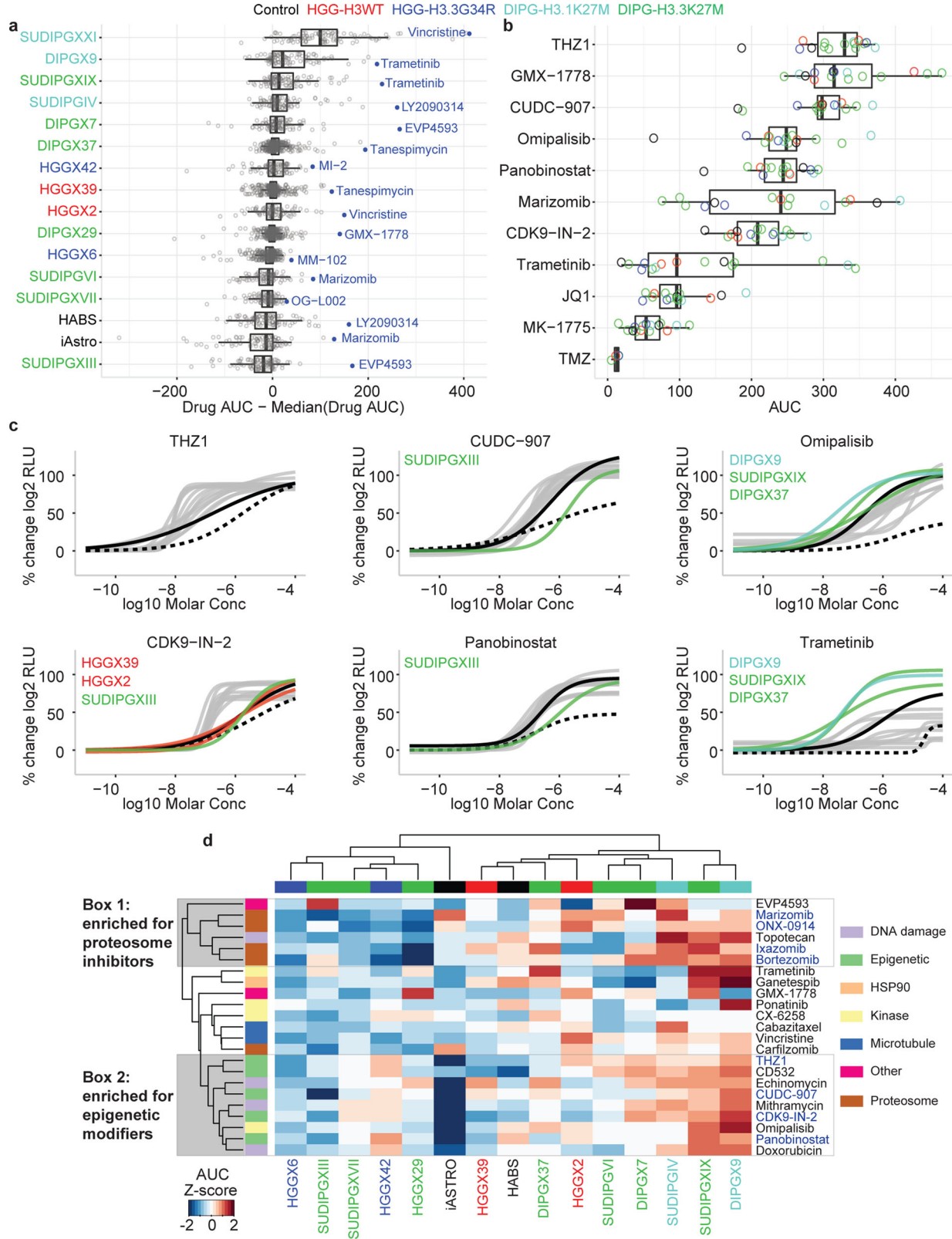

either drug alone, significantly extended survival of mice with intracranial SJ-DIPGX37 tumors (Fig. 9, $p = 0.0056$, Mantel–Cox log-rank test). We examined the plasma and brain pharmacokinetics of mirdametinib and paxalisib in normal mice to assess the possibility that the enhanced combinatorial effects were due to drug–drug interaction. While both agents had slightly increased plasma AUCs in combination (Supplementary Fig. 10 and Supplementary Data 5), the differences were within variability and less than twofold magnitude, the conventionally accepted threshold to identify drug–drug interactions[41]. Thus, the enhanced efficacy with combination treatment was likely due to combined pathway inhibition.

**Fig. 6 Analysis of screening results from 93 compounds across 14 pHGG models and two normal astrocyte lines. a** Distribution of normalized drug AUC (the drug AUC in that cell model minus the median AUC for that drug across all models to control for inherent drug potency). Each dot represents a single drug with the most selective drug for each model shown in blue. Cell models are color coded by histone mutation status. $n = 93$ drug AUC values derived from one or more independent experiments. **b** Distribution of AUC for select drugs that have been evaluated in clinical trials or implicated as promising agents in glioma preclinical studies. Each dot represents the drug AUC in one model and is color coded by the histone mutation status of the model. $n = 15$ drug AUC values derived from one or more independent experiments, except TMZ where $n = 4$. In **a**, **b**, data are represented as boxplots where the middle line is the median, the lower and upper hinges correspond to the first and third quartiles, the upper whisker extends from the hinge to the largest value no further than $1.5 \times$ the interquartile range (IQR) from the hinge and the lower whisker extends from the hinge to the smallest value at most $1.5 \times$ IQR of the hinge. All data points are plotted individually. **c** Select dose–response curves for the drugs highlighted in **b**. Normal references are depicted in black dashed lines (iAstro) and black solid lines (HABS). pHGG models are colored gray or by histone mutation status for models indicated. **d** Unsupervised hierarchical clustering of drug AUC z-scores for the 25% most active compounds out of 93 tested. Column and column labels are color coded by histone mutation status. Clusters 1 and 2 (gray boxes) highlight compound clusters showing distinct activity profiles across the models tested. The color code for histone mutation status is: H3-wt (red), H3.3 G34R (blue), H3.1 K27M (turquoise), and H3.3 K27M (green). Control cell lines (iAstro and HABS) are black. Color code for mechanism of action is shown on the right and annotated in heatmap row color blocks at left in **d**.

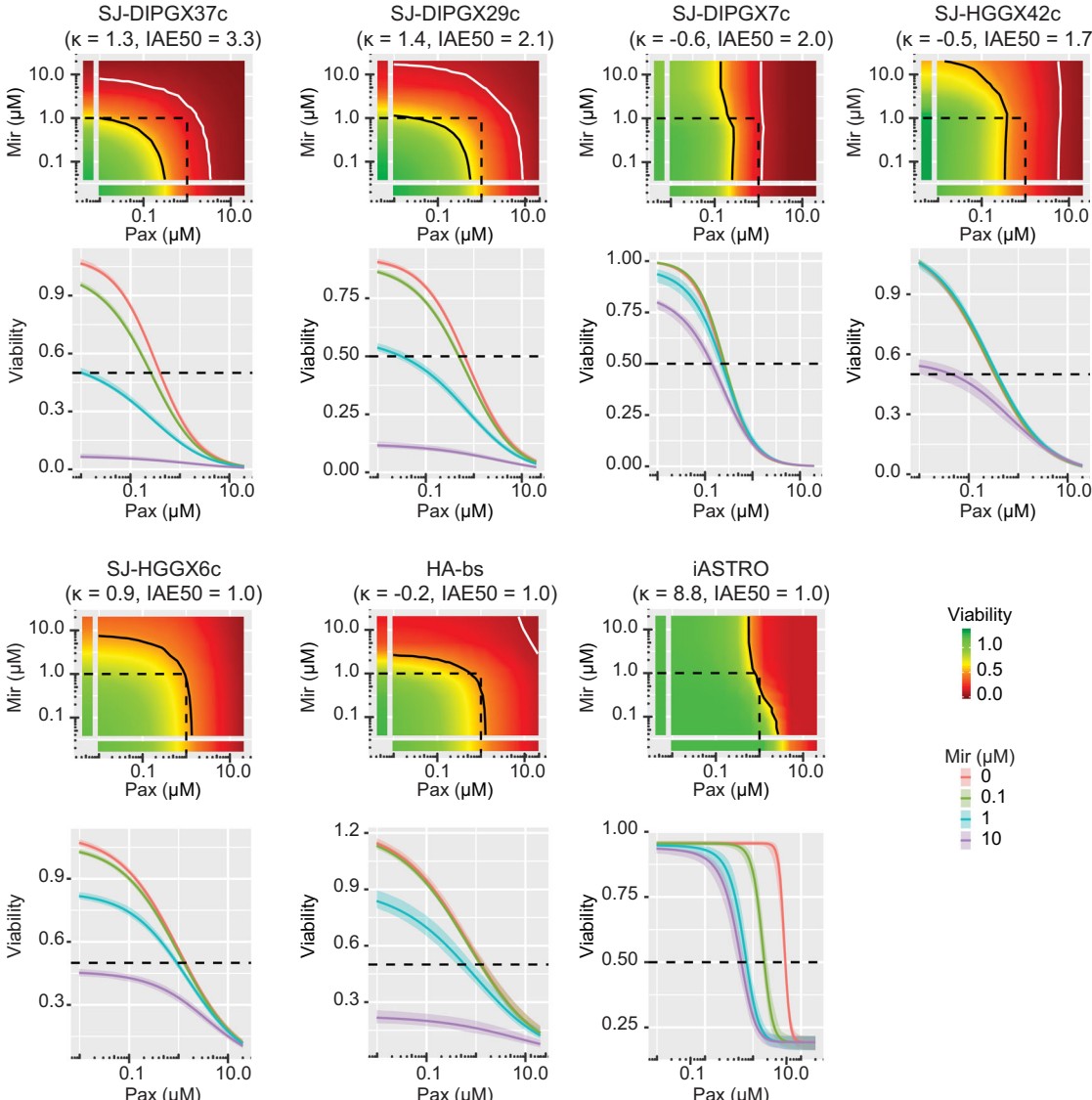

**Fig. 7 Paxalisib and mirdametinib drive synergistic growth inhibition in a subset of pHGG cell lines.** The BRAID model presents synergistic effects of paxalisib (pax) and mirdametinib (mir) following a 7-day treatment in 7-cell lines: H3.3 K27M DIPGs; SJ-DIPGX37c, SJ-DIPGX29c, SJ-DIPGX7c, H3.3 G34R pHGGs; SJ-HGGX42c, SJ-HGGX6c, and astrocyte controls HABS and iAstro. The parameter $\kappa$ measures the type of interaction: $\kappa < 0$ implies antagonism, $\kappa = 0$ implies additivity, and $\kappa > 0$ implies synergy). The index of achievable efficacy (IAE) quantifies the degree to which the drug combination achieves a minimal level of efficacy within a defined concentration range. Higher IAE means the combination was more efficacious. In this experiment, it was defined as a 50% reduction of cell viability (black line) at concentrations $\leq 1\,\mu M$ (dotted lines). The 90% reduction of viability isobole (white line) is included for reference.

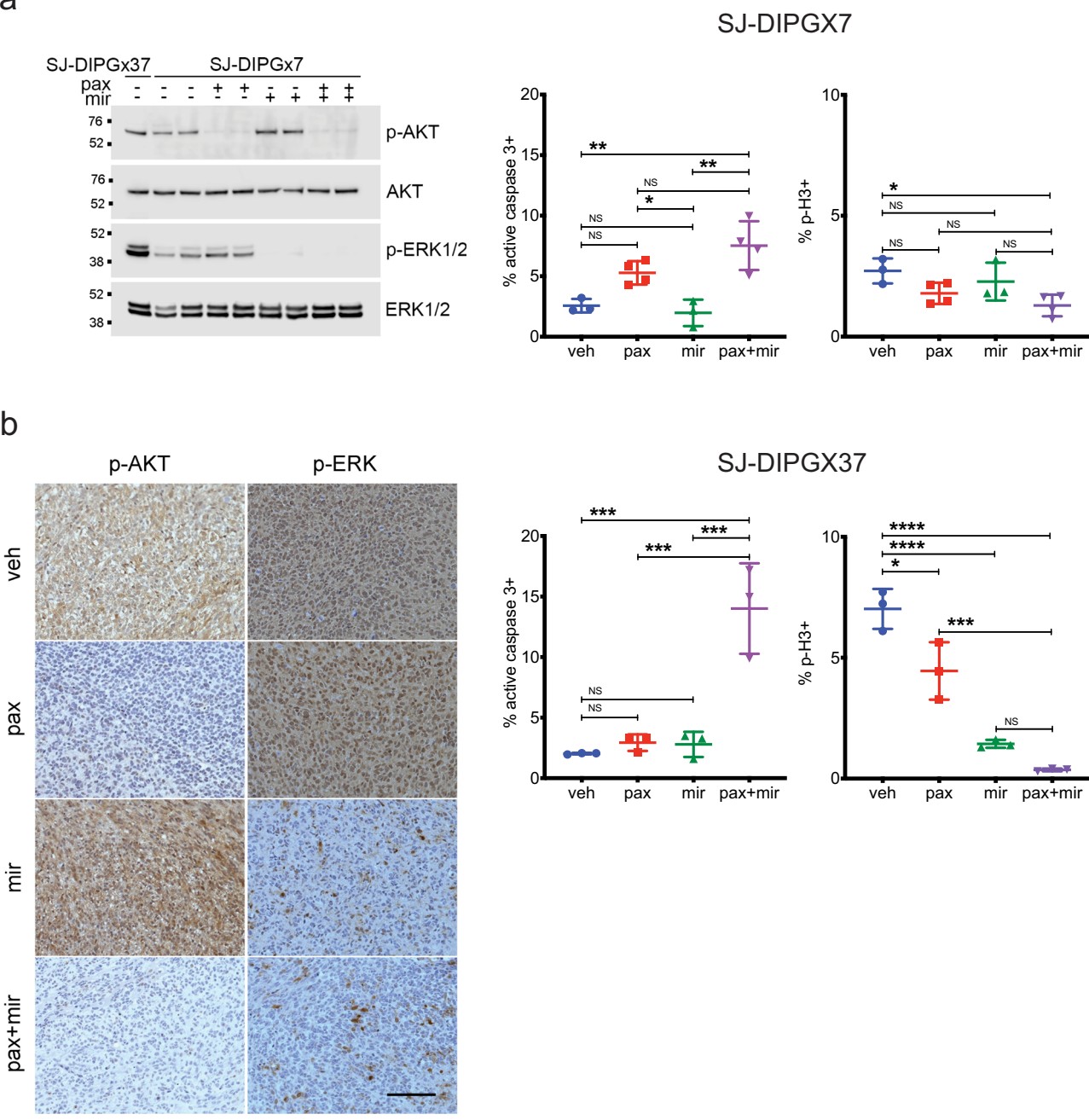

**Fig. 8 Paxalisib and mirdametinib show selective effects on cell survival and proliferation in vivo. a** Left: A single western blot of lysates from intracranial PDOX SJ-DIPGX37 (untreated, lane 1) and SJ-DIPGX7 (lanes 2–9) treated with vehicle (veh), paxalisib (pax, 12 mg/kg), mirdametinib (mir, 17 mg/kg), or the combination of paxalisib and mirdametinib (pax + mir) as indicated; antibodies are indicated at right. Quantification of IHC in sections from SJ-DIPGX7 tumors treated with agents shown along the *x*-axis for active caspase-3 (middle) and phospho-histone H3 (right), *n* = 3 tumors for veh, mir and *n* = 4 tumors for pax and pax + mir. The *p* values for comparison of active caspase-3 staining are veh vs. mir 0.9454, veh vs. pax 0.0964, veh vs. pax + mir 0.0032, mir vs. pax 0.0383, mir vs. pax + mir 0.0014, and pax vs. pax + mir 0.1474. The *p* values for comparison of p-H3 staining are veh vs. mir 0.7418, veh vs. pax 0.1733, veh vs. pax + mir 0.0258, mir vs. pax 0.6600, mir vs. pax + mir 0.1435, and pax vs. pax + mir 0.5747. **b** Left: IHC for pAKT Ser473 and pERK in SJ-DIPGX37 tumors in representative tumors treated with veh, pax, mir, or pax + mir as indicated. Quantification of IHC staining in sections from SJ-DIPGX37 tumors treated with agents shown along the *x*-axis for active caspase-3 (middle) and phospho-histone H3 (right), *n* = 3 tumors for each treatment. The *p* values for comparison of active caspase-3 staining are veh vs. mir 0.9628, veh vs. pax 0.9405, veh vs. pax + mir 0.0003, mir vs. pax 0.9997, mir vs. pax + mir 0.0005, and pax vs. pax + mir 0.0006. The *p* values for comparison of p-H3 staining are veh vs. mir < 0.0001, veh vs. pax 0.0110, veh vs. pax + mir < 0.0001, mir vs. pax 0.0043, mir vs. pax + mir 0.3352, and pax vs. pax + mir 0.0006. NS not significant; *$p$ < 0.05; **$p$ < 0.01; ***$p$ < 0.001, ****$p$ < 0.0001 using ordinary one-way ANOVA with post hoc Tukey test. The error bars indicate the mean ± s.d. Scale bar in **b**, left panel = 100 μm.

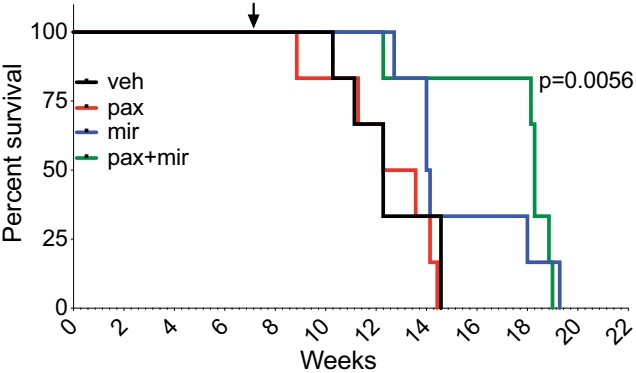

**Fig. 9 Combined treatment with paxalisib and mirdametinib significantly extends survival of SJ-DIPGX37-bearing mice.** Mice were randomized 50 days after implantation into four treatment arms (6 mice per arm) and treated with vehicle (black), paxalisib (8 mg/kg) (red), mirdametinib (14 mg/kg) (blue), or paxalisib + mirdametinib daily (green), 5 days ON and 2/3 days OFF. Paxalisib + mirdametinib vs. vehicle, $p = 0.0056$; mirdametinib vs. vehicle, $p = 0.16$; paxalisib vs. vehicle, $p = 0.48$ (Mantel–Cox log-rank test). Arrow shows the time point for randomization and initiation of treatment. Kaplan–Meier survival analysis.

## Discussion

pHGG remains largely incurable despite decades of clinical trials. Worldwide efforts in studying the molecular basis of this group of tumors has revolutionized understanding and revealed subgroups based on spatiotemporal tumor occurrence and striking molecular heterogeneity that has been further subdivided by DNA methylation signatures and inter- and intra-tumoral heterogeneity of mutated genes. Relevant in vitro and in vivo disease models that are founded in the correct developmental origins, recapitulate genetic and epigenetic signatures and represent the significant heterogeneity of pHGG are essential to further our understanding of mechanisms driving tumorigenesis and to identify therapeutic vulnerabilities. Although a growing number of DIPG cell lines have been established and characterized[15,19,42–44], relatively few of these efficiently engraft in the brain as xenografts for in vivo modeling[45], and there are a much smaller number of cell lines from pediatric gliomas arising outside the brainstem. PDOXs allow researchers to address critical dimensions of tumor biology, including angiogenesis, tumor invasion, and interactions with the tumor microenvironment, including the contribution of nervous system activity[46], that may strongly influence tumor growth and selective pressures. A recent study to establish a biobank of pediatric brain tumors reported the establishment of eight new pHGG PDOX models[47]. The 21 new pHGG PDOX models and 8 new cell lines reported here are a significant advance and include several rare tumor subtypes with limited available models, including three H3.3 G34R glioblastomas, three pHGG with MMRD, two glioblastomas in the pedRTKIII subgroup, and two PXAs. This new collection of models recapitulates the histopathologic and molecular hallmarks of pHGG and preserves genomic alterations (single-nucleotide variants, large-scale copy-number changes, and double minute chromosomes), mRNA expression profiles, and DNA methylome signatures of the primary tumors from which they were derived.

pHGG is known to display significant intra-tumoral heterogeneity due to clonal variation within the tumor[48–50]. Indeed, there were a few differences between patient tumor, PDOX, and cell line, such as *NTRK1* mutation detected in the autopsy sample of SJ-DIPGX37, but not the associated PDOX or cell line, or *MYCN* amplification present in SJ-DMGX40 PDOX, but not in the matched patient tumor (Fig. 4 and Supplementary Fig. 2). Such differences could be due to regional heterogeneity

between the patient sample analyzed and cells implanted, expansion of a minor subclone during PDOX establishment, or acquisition of new mutations during ongoing tumor evolution in the mouse host.

Across multiple PDOX models, we found consistent upregulation in genes associated with proliferation and decreased expression of genes associated with inflammatory response compared to matched patient tumors, as previously described[47]. After removing these shared PDOX-specific signatures, however, there was a strong correlation in expression signatures between PDOX and matched patient tumors. Expression differences between the patient tumors and derived PDOX may be due to greater tumor purity in the PDOX expression profiles after removing mouse reads from non-tumor cells within the sample. However, the process of establishing cell lines or PDOXs may select for more proliferative populations. While such selection could lead to an overestimate of the effects of cell cycle inhibition in pHGG models, such bias was not observed in the control cell lines that proliferated more rapidly than the tumor cells.

Despite decades of clinical trials, no effective chemotherapy approaches have been identified for pHGG. However, the revolution in knowledge gained through genomic analyses of pHGG revealed some clear therapeutic targets, and selective inhibitors have induced striking responses in pHGG with *BRAF* V600E mutation or *NTRK* fusion genes. The high frequency of H3K27M mutations in DIPG and other midline gliomas sparked an intense search for selective vulnerabilities conferred by this mutation, especially in agents connected to epigenetic regulation. Some of the top hits from our HTS were consistent with previous findings in DIPG cell line screening, including inhibitors of HDACs, CDK7, and CDK9, and proteins involved in epigenetic or transcription regulation[15,19,24,28,51]. The inclusion of additional pHGG models in our experiments showed the broad efficacy of these agents independent of H3K27M mutation. The 7-day assay performed in our study could underestimate the efficacy of some epigenetic modulatory compounds, which may take longer to manifest than compounds inducing acute cytostatic or cytotoxic responses in growth inhibitory assays.

Overall, the two sources of normal astrocyte control cells were among the least sensitive cell lines to the collection of compounds tested in DR (Fig. 6a), and many of the top hits showed greater sensitivity in tumor cell lines than in iAstro control cells (Fig. 6b, c). However, responses varied between the two sources of astrocytes (Fig. 6b, c and Supplementary Fig. 8). These results reveal potential differences in drug sensitivity in normal cells at different developmental stages as well as variation among tumors, and further highlight the difficulty in determining simple predictors of drug responsiveness in pHGG which are heterogenous in developmental expression signatures as well as genetic mutations.

Understanding how disease heterogeneity contributes to therapeutic response is the foundation of precision medicine. The panel of 14 pHGG cell lines and two astrocyte controls revealed substantial variation in efficacy and potency across the collection of compounds representing multiple MOA. We further investigated inhibitors of PI3K/mTOR and MEK pathways in two PDOX models with differing responses as test cases to determine how well results of in vitro screening were predictive of in vivo response. Myriad mutations in pHGG and other cancers converge on these two signaling pathways, and there are multiple examples of pathway cross-talk that compromise the efficacy of single-agent approaches that inhibit only one of these two regulatory cascades[38,39,52,53]. Among available inhibitors of PI3K/mTOR and MEK, we selected paxalisib and mirdametinib, respectively, for in vivo studies because of their ability to traverse the blood–brain barrier. In vivo testing with these drugs induced dramatic inhibition of pathway activity in both PDOXs with

differential tumor responses consistent with in vitro studies. SJ-DIPGX7 showed relative resistance to MEK inhibition and a more moderate impact of combined PI3K/mTOR inhibition compared with the more responsive SJ-DIPGX37, where the drug combination drove greater cytotoxic and cytostatic responses and extended survival of mice carrying intracranial tumors. Additional in vivo studies would be needed to determine if the in vitro screening results are predictive of response in all cases. Identifying reliable biomarkers to accurately predict response from biopsy samples will likely require a significantly larger collection models. Pharmacokinetic (PK) analyses showed no substantial drug–drug interaction changing drug exposures, so the enhanced effects are likely to represent the result of combined pathway inhibition. However, levels of mirdametinib used to effectively block signaling in our study were significantly higher than those observed in humans following exposure at the current clinical dosing of 4 mg twice daily[54]. Given the compelling survival advantage and observed in vivo toxicity, there is a rationale to consider local delivery of PI3K/mTOR and MEK inhibitors to these tumors to avoid systemic toxicities and reach the necessary levels of drug activity.

It is promising that the majority of compounds screened in our study showed substantially greater efficacy across all of the tumor models tested when compared to temozolomide, which is still frequently used in treating pHGG. Further studies are needed to understand the intrinsic differences in sensitivity between tumors to ultimately achieve the promise of personalized medicine. While we did not observe a simple predictive relationship between DNA methylation subgroup and compounds inhibiting a particular MOA, our results showed efficacy of inhibitors of transcription and translation across all pHGG subgroups (Fig. 6b, c). Importantly, we also found that each cell line showed selective responses to specific compounds (Fig. 6a) that were not predicted by simple correlation with DNA methylation subgroups or mutations in signature pHGG genes, indicating that there are clear but heterogeneous vulnerabilities for these tumors. Thus, major advances in the treatment of this heterogeneous group of intractable brain tumors will require significant new insights into mechanisms driving disease pathogenesis and even more extensive preclinical testing to identify more reliable predictors of these selective vulnerabilities. Models with a detailed molecular characterization that can be experimentally manipulated and studied in vitro and in vivo provide powerful tools needed to address this challenge. To facilitate an in-depth exploration of this collection of 21 new PDOX and 8 new cell lines, the Pediatric Brain Tumor Portal provides an interactive interface to access and explore all clinical and molecular data and drug screening results. It provides summary overviews for each model (Supplementary Fig. 5) and tools to generate customized mutation oncoprints, gene expression heatmaps, and overlays of dose–response curves for selected drugs and cell lines. The well-characterized models reported here provide a rich resource for the pediatric brain tumor community.

## Methods

*Patient material*: Genomic analyses of patient material and use of patient tumor samples to establish xenografts and cell lines were performed with informed consent and approval from the Institutional Review Board of St. Jude Children's Research Hospital. Written informed consent was obtained from patients and/or legal guardians for use of tissue for research. This study complies with the Declaration of Helsinki and all other relevant ethical regulations.

All reagents used for this study are listed in Supplementary Table 2.

**PDOX establishment**. Surgical or autopsy tumor samples were transported in neurobasal or DMEM/F12 media without additives at 4 °C. Most tissues were processed directly, although samples stored overnight at 4 °C also successfully engrafted to form tumors. Tissues were dissociated into a single-cell suspension by gentle pipetting in warm neurobasal media or by enzymatic dissociation with papain. Single tumor cells were isolated by a mixture of 1% activated Papain in

Neurobasal™ media with 0.16 mg/ml N-acetyl cysteine and 0.012 mg of DNase through a 15–30 min digestion at 37 °C. The cells were washed in Neurobasal™ media, counted, and resuspended in Matrigel (Corning, Tewksbury MA, US). Intracranial implantation of $2 \times 10^5$–$1 \times 10^6$ cells in matrigel into CD-1 nude mice was completed as described[55]. In brief, intracranial injections for tumor implantations were performed by designated surgeons, under general anesthesia and with analgesics for 1 week. The head of the anesthetized mouse (female CD-1® Nude Mouse, Charles River, Wilmington, MA, US) was held by ear bars on a stereotaxic equipment (David Kopf Instruments Tujunga, CA, US). A standard $2 \times 2$ mm bore hole was created with a dental drill and the dura was cut completely from the surface of the exposed brain. Cells in Matrigel were injected with a sterile glass Hamilton syringe bearing a 26G unbeveled needle 2.5 mm below the cortex surface. The wound was sealed with two Relex clips (Kent Scientific Corporation, Torrington, CT, US). Mice were monitored until awake. The drinking water contained 115 µg/ml ibuprofen and Baytril (3.5 ml of 2.27% stock/500 ml water) for a week. Mice were observed and monitored daily for neurological and health symptoms and euthanized at a humane endpoint. PDOX tumors were dissected from moribund mice, dissociated, and passaged into 5–10 recipient mice, or cryopreserved in either Millipore or Sigma cell freezing medium (Supplementary Table 2). Models were considered established after successfully engrafting through three passages. Recovery from cryopreserved cells typically showed delayed in vivo tumor growth of 1.5–2 times compared to passaging from fresh tumors. Many lines were transduced with a lentivirus to express luciferase and yellow fluorescent protein (CL20-luc2aYFP) for in vivo imaging. Mice were maintained in an accredited facility of the Association for Assessment of Laboratory Animal Care in accordance with NIH guidelines. Housing conditions for mice were temperature range 68–74 °F, humidity range 30–70%, dark/light cycle cycle 12 h dark/12 h light with lights on at 6 AM, off at 6 PM. The Institutional Animal Care and Use Committee of SJCRH approved all procedures in this study.

**Cell line propagation and maintenance**. To establish pHGG cell lines SJ-HGGX2c, SJ-HGGX6c, SJ-HGGX42c, SJ-HGGX39c, SJ-DIPGX7c, SJ-DIPGX9c, SJ-DIPGX29c, and SJ-DIPGX37c, fresh PDOX tumors were dissected from euthanized mice, and sterile tumor tissue was digested to single cells as for PDOX establishment above. The cells were washed and plated in Corning® Ultra-low Attachment plates in media used for neural stem cells and glial progenitor cells consisting of a 1:1 mixture of Neurobasal™ without phenol red (with 2% of B27 without vitamin A and 1% of N2) and ThermoFisher Knock-Out DMEM/F12 (with 2% of Stempro® neural supplement) supplemented with 20 ng/ml of human recombinant EGF, 20 ng/ml of human recombinant bFGF, 10 ng/ml of human recombinant PDGF-AA and -BB, 1% of Glutamax, 1% of sodium pyruvate, 1% of NEAA, 10 mM of HEPES, 2 µg/ml of heparin and 1× Primocin. In the first passage, the Miltenyi Biotec Mouse Cell Depletion Kit was used, and removal of residual mouse cells was verified by demonstrating successful PCR amplification with humans, but not mouse-specific, primer sets for *H3F3A/h3f3a* (Supplementary Table 3). Sequences of PCR products were verified. To further verify that cultured cells were of human origin, immunofluorescent staining with anti-human mitochondria and anti-human nuclear antigen antibodies was performed.

The cell lines were either maintained in suspension culture as tumorspheres or on human ESC-qualified Geltrex artificial extracellular matrix-coated tissue culture surface[56] at 37 °C, 5% $CO_2$, and 5% of $O_2$. Doubling times were computed using http://www.doubling-time.com/compute.php.

SU-DIPG-IV, SU-DIPG-VI, SU-DIPG-XIII, SU-DIPG-XVII, SUDIPG-XIX, and SU-DIPG-XXI[15,19,20] were generous gifts from Dr. Michelle Monje. Normal cell type references were human neural stem cells induced from H9 ES cells (Invitrogen, N7800-100), human iPSC-derived astrocytes (Tempo Bioscience), and human brainstem astrocytes (ScienCell Research Laboratories, #1840).

**Short tandem repeat profiling**. Molecular fingerprinting for PDOXs and cell lines was performed with Promega PowerPlex® 16 HS or PowerPlex Fusion® System (Promega Corporation, Madison, WI).

**DNA methylation profiling and copy-number analysis**. DNA methylation profiles were evaluated using Illumina Infinium Methylation EPIC BeadChip arrays according to the manufacturer's instructions. Raw IDAT files from pHGG patient tumors, PDOXs, and cell lines as well as a published reference cohort from ref. [12] were assessed for quality control and pre-processed using the minfi package in R[57]. Low-quality samples were excluded from the downstream analysis based on mean probe detection with $p$ value > 0.01. Relevant tumor subgroups were selected from the reference cohort. Subgroups with more than 30 cases were randomly down-sampled to 30 cases. Methylation probes were filtered based on the following criteria: detection $p > 0.01$ in >50% of the cohort, non-specificity based on a published list from ref. [58], and probes residing on sex chromosomes. The data then underwent single-sample Noob normalization to derive beta values, with the top 35,060 most variable probes (probe s.d. > 0.25) selected for downstream analysis. Distances between samples were calculated based on Pearson's distance and then visualized with the t-SNE algorithm (Rtsne package v.0.11).

Copy-number alteration (CNA) analysis was done using the conumee package (v 1.18.0) with default parameters. Probe intensities were normalized against a

reference set of normal brain tissues profiled by Methylation EPIC array ($n = 34$). CNAs were then detected as significant positive or negative deviations, which encompassed more than 50% of the chromosomal arm, from the genomic baseline.

**Neuropathology assessment.** Standard hematoxylin and eosin histopathologic preparations of 5-μm formalin-fixed and paraffin-embedded tissue sections from patient tumors and derived PDOXs were centrally reviewed by a board-certified neuropathologist specialized in pediatric CNS tumors (J.C.) and blinded to the origin and genetic alterations of the PDOXs.

**Immunohistochemistry and FISH.** Immunohistochemistry (IHC) was performed as previously described[59]. In brief, tissues were fixed in 4% paraformaldehyde at pH 7.4, embedded in paraffin, and cut into 5 μm sections. For DAB staining, antigen retrieval was performed by boiling in 0.01 M citrate buffer (pH 6.0) for 10 min. Endogenous hydrogen peroxidase activity was blocked by 0.6% hydrogen peroxide in TBS (pH 7.6) for 30 min, and IHC followed instructions from the VECTAS-TAIN Elite ABC kit. Antibodies and dilution factors are listed below and in Supplementary Table 2. Amplification of *PDGFRA* (4q12) and *MYCN* (2p24) was detected by interphase fluorescence in situ hybridization in a Clinical Laboratory Improvement Amendments (CLIA)-certified laboratory in St. Jude Children's Research Hospital with probes developed in-house using the following BAC clones: *PDGFRA* (RP11-231C18 + RP11-601I15) with 4p12 control (CTD-2057N12 + CTD-2588A19) and MYCN (RP11-355H10 + RP11-348M12) with 2q35 control (RP11-296A19 + RP11-384O8).

**Antibodies.** Primary antibodies used were Akt rabbit pAb, dilution ratio 1:1000 for Western (Cell Signaling, 9272); Phospho-Akt (Ser473) rabbit pAb, dilution ratio 1:1000 for Western, 1:50 for IHC (Cell Signaling, 9271); p44/42 MAPK (Erk1/2) rabbit pAb, dilution ratio 1:1000 for Western (Cell Signaling, 9102); Phospho-p44/42 MAPK (Erk1/2) (Thr202/Tyr204) rabbit pAb, dilution ratio 1:1000 for IHC and Western (Cell Signaling, 9101); Phospho-Histone H3 (Ser10) rabbit pAb, dilution ratio 1:200 for IHC (Cell Signaling, 9701); Cleaved Caspase-3 Antibody rabbit mAb, dilution ratio 1:500 for IHC (BD Pharmingen™, #559565); and Anti-ATRX rabbit pAb, dilution ratio 1:600 for IHC (Millipore Sigma, HPA001906). The secondary antibodies were anti-rabbit biotinylated secondary antibody, dilution ratio 1:200 for IHC (Vector Laboratories, BA-1000); Rabbit IgG HRP linked whole Ab, dilution ratio 1:10,000 for Western (MilliporeSigma GENA934-100UL); and Mouse IgG HRP-linked whole Ab Cytiva, NXA931, dilution ratio 1:10,000 for Western (MilliporeSigma GENXA931-1ML).

**WGS and WES analysis.** Both WGS and WES were performed on the majority of the samples with a few samples having only WGS or WES. Paired-end sequencing was conducted on Illumina HiSeq platform with a 100- or 125-bp read length or NovaSeq with 150-bp read length. Paired-end reads from WGS and WES were mapped to GRCh37-lite using BWA-aln followed by QC[60–62]. For PDOX samples, mapped reads were cleansed of mouse read contamination by XenoCP[63]. For patient tumors and PDOX samples with matched germline samples, somatic mutations including SNVs and Indels were called and classified[61,62]. Briefly, the putative somatic calls were called with Bambino in the paired tumor/normal mode and normal only modes, then further filtered to ensure the high sequence level supports including (1) strong enrichment in tumor sample compared with normal samples, using Fisher exact test; (2) supporting reads in tumors in both orientations of mated pairs (≥3 unique reads); (3) low read allele frequency in normal samples (≤0.05). Additional filters are included to remove the somatic mutations due to the paralogous mapping, sequence misalignment, or sequencing artifacts due to homopolymers, or low-quality base pairs. Non-silent mutations including missense, nonsense, in-frame insertion, in-frame deletion, frameshift, and splice mutations were reported. Potentially pathogenic germline variants were reported based on these filters: (1) non-synonymous mutations with variant allele frequency (VAF) > 0.2 and coverage > 10×; (2) minor allele frequency (MAF) in general populations <1e−3 in ExAC;[64] and (3) REVEL score > 0.5 (ref. [65]), if available, for missense mutations. For patient tumors and PDOX samples without matched germline samples, variants were called and annotated by Bambino and Medal Ceremony[66,67]; variants meeting these criteria were reported as potentially pathogenic variants: (1) any "Gold" variants annotated by Medal Ceremony with alternative allele count > 4; (2) non-Gold but non-silent variants with VAF > 0.3, alternative allele count > 4, and MAF in general populations < 1e−3 in ExAC[64], 1000 Genomes, and NHLBI. Mutations in pHGG signature gene list (Supplementary Data 2b) were manually reviewed. Oncoprints were created using the online tool ProteinPaint[68]. CNAs were called and annotated by CONSERTING[69], and focal CNA covering signature genes were manually reviewed.

**RNA-sequencing analysis.** Paired-end RNA sequencing (RNA-seq) was conducted on Illumina HiSeq platform with 100-bp read length or 125-bp read length. Paired-end reads from RNA-seq were mapped by BWA and STAR to multiple reference files such that the best alignments are selected to be included in the bam file[70]. For PDOX samples, mapped reads were cleansed of mouse read contamination by XenoCP. Read counts per gene per sample were quantified by HTSeq-Count v 0.11.2 using level 1 and level 2 transcripts of GENCODE v19

annotation[70]. Expression heatmap was plotted based on log CPM of genes from three expression signatures across the entire cohort, excluding four samples due to different RNA-seq protocol (patient diagnostic and recurrent samples from SJ-HGGX2, patient and PDOX samples from SJ-HGGX6) and four PDOX samples from earlier passages (one PDOX from SJ-DIPGX29, one PDOX from SJ-HGGX39, and two PDOXs from SJ-DIPGX7). Differential expression analyses were carried out on 16 one-to-one matched pairs of PDOXs from the most recent passages and the patient samples from which they were derived using edgeR (v 3.28.0) and limma (v 3.42.0) following the RNAseq123 workflow[71]. Subsequent gene set enrichment analyses were conducted using a hypergeometric test against hallmark gene sets downloaded from MSigDB v 5.2 (ref. [72]). Fusion genes were detected using CICERO[73].

**High-throughput screening.** Growth curves for each cell line were established in 384-well plates (Corning 3707 or 3765) coated with 40 μl of 1% Geltrex matrix with varying cell numbers to determine optimal seeding density for 7-day treatments (Supplementary Table 1). For HTS, all assays were performed with the negative control (DMSO, 0.35%) and the positive control (staurosporine, 19–35 μM) in parallel. The FDA single-point assays were performed at a final concentration of 33 μM [95% CI 16–46 μM]. Dose–response experiments were performed with 10-point, threefold serial dilution (19683-fold concentration range), and the mean top concentration tested was 35 μM [95% CI 16–50 μM] (Supplementary Data 4).

The assay plates were loaded into an automated cell culture compatible LiCONiC incubator (37 °C, 5% $CO_2$ and humidified) (LiCONiC US, Woburn, MA) that was integrated into an automated HTS system (HighRes Biosolutions, Beverly, MA). All compounds were transferred with a pin tool (V&P Scientific, San Diego, CA) and tested in triplicate, and drug exposure time was 7 days except for the fast-growing iAstro cells (shortened to 3 days). Media and drug was not changed during the 7-day incubation to avoid logistical difficulties including cell loss with media change. At the end of the experiment, the assay plates were equilibrated to room temperature for 20 min. Cell viability was measured using CellTiter-Glo® (Promega, Madison, WI). Luminescent signal was read with an EnVision® multimode plate reader (PerkinElmer, Waltham, MA). Screening experiments were processed, and the results visualized using two in-house developed programs: RISE (Robust Investigation of Screening Experiments) and AssayExplorer.

**Data analysis of drug responses.** Raw data processing—log2 RLU dose–response fits

Raw luminescence relative light unit (RLU) values for each compound at each concentration were log2 transformed, normalized to obtain % activity using the following equation: $100 \times [(\text{mean}(\text{negctrl}) - \text{compound})/(\text{mean}(\text{negctrl}) - \text{mean}(\text{posctrl}))]$, and then pooled from replicate experiments prior to fitting. Here, negctrl and posctrl refer to the negative (DMSO) and positive controls (staurosporine) on each plate.

Dose–response curves were fit using the *drc*[74] package in R [R Core Team (2012). R: A language and environment for statistical computing. R Foundation for Statistical Computing, Vienna, Austria. ISBN 3-900051-07-0, URL http://www.R-project.org/]. Both a three-parameter (with $y_0$, the response without drug, set to zero) and a four-parameter model ($y_0$ allowed to vary) were fit using the sigmoidal function *LL2.4*. The hill slope was constrained to be between −10 and 0, and $EC_{50}$ was constrained to be between $10^{-11}$ and $10^{-4}$ (which roughly equated to the drug concentration range tested in these experiments). For the three-parameter model, $y_{Fin}$, the maximum response of the dose–response curve, was constrained to be between zero and the maximum of the median activities calculated at each concentration overall pooled measurements. For the four-parameter model, $y_0$ and $y_{Fin}$ were both constrained to be between the minimum and the maximum of the median activities calculated at each concentration overall pooled measurements. The model with the lowest corrected Akaike information criterion (AICc) was selected as the best fit model.

Area under the curve (AUC) was calculated from the fitted curve using the trapezoid rule in the concentration range $10^{-11}$ and $10^{-4}$ M. In the event of a failure to fit a sigmoidal dose–response curve, the *smooth.spline* option in R was used to fit a curve that could be used to determine AUC.

**QC metrics for HTS.** The z′ statistic was calculated using the following formula: $1 - ((3 \times \text{s.d. (negctrl)}) + (3 \times \text{s.d. (posctrl)}))/\text{abs (mean (negctrl)} - \text{mean (posctrl))}$.

**BRAID model for quantitative synergy analysis.** Drug combination experiments were analyzed using the BRAID response surface model[40]. Raw RLU values were processed as described above for the single-agent experiments, except no log2 transformation was applied.

**In vivo testing of paxalisib and mirdametinib.** Paxalisib (GDC0084) and mirdametinib (PD0325901) were formulated at 1.8 or 2.5 mg/ml, respectively, in 1% methylcellulose and 1% Tween 80 with sonication. The combination was co-formulated and administered as a single gavage.

**Pharmacodynamic assays**. SJ-DIPGX37 or SJ-DIPGX7 tumors were implanted in the brain and treated when mice showed decreased activity consistent with large tumors and confirmed visualization of brain tumors by MRI. Mice were dosed by oral gavage, QD, for 5 days with mirdametinib (17 mg/kg), paxalisib (12 mg/kg), mirdametinib and paxalisib (17 mg/kg and 12 mg/kg, respectively), or vehicle. Two hours after the last dose, mice were perfused with phosphate-buffered saline (PBS) to remove blood, a small piece of tumor was grossly dissected and snap-frozen for western blot analysis, and the remainder of the tumor and brain was processed for FFPE histology and IHC. Long-term treatment at these doses caused toxicity manifested as loss of >20% of body weight. To identify doses tolerated for longer treatment needed for survival studies, we treated CD-1 nude mice without tumors in one of seven arms: (1) vehicle; (2) paxalisib at 8 mg/kg; (3) paxalisib at 12 mg/kg; (4) mirdametinib at 14 mg/kg; (5) mirdametinib at 17 mg/kg; (6) paxalisib at 8 mg/kg + mirdametinib at 14 mg/kg, or (7) paxalisib at 12 mg/kg and mirdametinib at 17 mg/kg. Three mice per arm were treated for five consecutive days, and then the brains were collected 2 h after the final dose for pharmacodynamic analysis of pathway inhibition (pAKT and pERK) by western blotting. An additional three mice per arm were treated on cycles of 5 days on, 2 days off for three cycles. Body weight and behavioral changes were monitored daily. Paxalisib (8 mg/kg) and mirdametinib (14 mg/kg) alone and in combination were tolerated without loss of >20% body weight, so these doses were selected for a survival study. Western blots for pharmacodynamic studies were performed as previously described[59]. In brief, the tumor tissue was homogenized in cold RIPA buffer with protease and phosphatase inhibitor cocktails on ice. The protein concentrations were determined with the BCA method. The samples were denatured with the NuPAGE sample-reducing agent and LDS sample buffer at 70 °C for 15 min. The proteins were separated by NuPAGE precast gels and blotted on the nitrocellulose membrane. Antibodies including the dilution ratios and reagents are listed in the key resource Supplementary Table 3. The western blot was imaged with Li-Cor Odyssey® Fc. Uncropped and unprocessed scans of western blots are available in the Source Data Supplementary File.

**Survival study**. Cryopreserved SJ-DIPGX37 PDOX cells were thawed, and $2.4 \times 10^5$ cells in 7.5 μl of Matrigel/mouse were implanted into brains of 24 mice. The details of implantation were previously described[55] in this Methods section and in a previous publication[55]. Bioluminescence imaging (BLI) was monitored weekly, and 50 days after implantation when all mice reached a threshold BLI total flux $>2 \times 10^5$, they were randomized into four treatment groups (6 mice per group) and treated with vehicle, paxalisib (8 mg/kg), mirdametinib (14 mg/kg) or paxalisib + mirdametinib QD, 5 days on and 2/3 days off. Mice were euthanized when they reached moribund status.

**PK study and analysis**. The plasma and brain PK of mirdametinib and paxalisib was studied in non-tumor-bearing mice to determine potential for a PK drug–drug interaction (DDI). Mice were dosed mirdametinib and paxalisib daily, either alone or in combination, for up to 5 days. Blood samples were obtained from the retro-orbital plexus under anesthesia or by cardiac puncture upon termination. Brains were harvested after cardiac puncture and aortic perfusion with PBS. Samples were stored at −80 °C until analysis with qualified LC-MS/MS methods. Plasma concentration-time (Ct) data were analyzed using nonlinear mixed effects modeling implemented in Monolix 2019R2 (Lixoft, Antony, France). To enhance modeling precision and power, additional paxalisib plasma Ct data from a separate DDI study with another targeted compound was added to the analysis. Combination status was tested as a covariate upon the apparent clearance (CL/F) of each compound using the likelihood ratio test. Brain concentrations were log transformed and analyzed using two-way ANOVA with sample time and combination status as factors. Practically impactful interactions were defined as a ≥twofold change in the parameters of interest, consistent with the conventionally accepted threshold for preclinical and DDI studies[41,75]. Additional details are presented in Supplementary Note 1.

**Reporting summary**. Further information on research design is available in the Nature Research Reporting Summary linked to this article.

## Data availability

DNA methylation profiling data generated in this study are available in the Gene Expression Omnibus (GEO) database, accession GSE152035. Next-gen sequencing data generated in this study have been deposited at the European Genome-Phenome Archive (EGA), which is hosted by the European Bioinformatics Institute (EBI), and are available under the indicated accession numbers for whole-genome (EGAS00001005159), whole exome (EGAS00001005160) and RNA sequencing (EGAS00001005161). Gene set enrichment analyses used hallmark gene sets from MSIGDB v 5.2 [http://www.gsea-msigdb.org/gsea/msigdb/index.jsp]. Interactive visualizations of data can be explored in the Pediatric Brain Tumor Portal (pbtp.stjude.cloud). A reporting summary for this article is available as a Supplementary Information file. The main data supporting the findings of this study are available within the article and the Supplementary Figures, Tables and Data. The source data underlying Figs. 8 and 9 and Supplementary Figs. 8b, 9a–c, 10a–d are provided in a Supplementary Source Data file. Source data are provided with this paper.

## Materials availability

Xenografts and cell lines generated for this study can be requested through the Pediatric Brain Tumor Portal (pbtp.stjude.cloud), and with completion of a material transfer agreement. Limited additional patient information is available from the authors upon request. Source data are provided with this paper.

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

## Acknowledgements

We are grateful to the patients and families who donated tissue to support pHGG research. This work was funded in part by the National Brain Tumor Society Defeat Pediatric Brain Tumors Research Collaborative (to S.J.B., C.T., and A.A.S.), NIH CA096832 (to S.J.B., M.F.R., and J.Z.), the NCI Cancer Center Support Grant CA21765, Musicians Against Childhood Cancer, and ALSAC. We thank the St. Jude Children's Research Hospital (SJCRH) Research Information Services and Cloud Applications groups for the development of the online Pediatric Brain Tumor Portal, Biomedical Communications for Fig. 1 artwork, Center for In Vivo Imaging and Therapeutics (CIVIT), Hartwell Center, and Biorepository. We thank Drs. Jie Zhang and Ali G. Saad (Le Bonheur Children's Hospital) for fresh tissue collection, Asli Goktug (SJCRH) for assistance with HTS, Brittney Gordon in the SJCRH DNB Xenograft Core for assistance with PDOX cryopreservation, and Dr. Arzu Onar-Thomas for assistance with statistical analyses.

## Author contributions

C.H., K.X., X.Z., P.S.D., J.C., C.L.T., A.A.S., and S.J.B. conceived and designed the project. X. Z., C.H., C-H.K., K.S.M., and M.F.R. established or provided PDOX models; C.H. and M.M. established cell lines; K.X., B.G., J.Z., P.A.N., and G.W. performed genome, transcriptome, or methylome computational analyses; J.C. and B.A.O. performed histopathological evaluations; C.H., W.L., N.T., X.L., B.J., D.G.C., J.X., T.C., Z.R., C.L.T., A.A.S., and S.J.B. performed or designed drug/compound testing experiments; A.A.S. and N.T. analyzed compound screening data; P.S.D., L.D.H., L.H.K., J.Z., X.L., and J.D. provided experimental support and coordination of samples and data; W.C., Y.W., and B.B.F performed pharmacokinetic analyses; C.L.T., A.B., C.W., S.A.U., I.Q., P.K., F.B., and A.G. contributed to identification and provision of clinical samples; C.L.T. and J.C. provided clinical annotations; P.S. D., A.A.S., and S.J.B. coordinated with IT for design and development of the Pediatric Brain Tumor Portal; C.H., K.X., J.C., C.L.T., A.A.S., and S.J.B. wrote the manuscripts with feedback from all the authors; J.C., C.L.T., A.A.S., and S.J.B. jointly supervised this work.

## Competing interests

The authors declare no competing interests.
