## [Peer Review File · Nature Communications]

REVIEWER COMMENTS

Reviewer #1 (Remarks to the Author):

This manuscript describes the development of a panel of patient-derived orthotopic xenografts (PDOXs) and cell lines from pediatric high-grade glioma. The authors have subjected these models to histological and molecular analysis, including DNA methylation profiling, whole genome or whole exome DNA sequencing and RNA sequencing, and have made all the data available through an online portal so they can benefit the research community. In addition, they have performed high-throughput drug screening on a subset of the cell lines, as well as on cell lines acquired from other institutions. The drug screening reveals variability in drug responses across different patient-derived models, and identifies individual drugs and drug combinations that may be effective for therapy. Finally, some of these drugs are tested in vivo, and shown to have an impact on tumor growth and on animal survival.

The studies described here represent an extraordinary effort that will be of significant benefit to the field of pediatric oncology. The generation of PDOXs and cell lines, the detailed characterization of these lines, and the database that shares these resources with the community, will certainly help move the field as a whole forward. In addition to creating these resources, the authors use them to interrogate drug responses, which is an important effort that could yield novel therapies for diseases that sorely need them. But while this is an important and timely study, there are a number of issues with the presentation and analysis of data that need to be addressed before the paper is suitable for publication.

1) The supplementary tables present important information, but in many cases they are hard to follow. Each table should have a legend incorporated at the top, instead of just in a separate Supplementary Information file. The legend should clearly define the abbreviations used in each of the column headings so that the reader can follow what is being presented. Supp Table 5 is particularly challenging; even with the information provided in the Supplementary info file it is hard to know what each of the tabs are, and to interpret the contents of those tabs. As another example, Supp Table 7 has multiple abbreviations that are not defined. Making all of the tables more user-friendly is essential to allow readers to understand and benefit from the authors' work.

2) Typically, methylation analysis provides a score denoting the similarity between a given sample and the reference cohorts for each tumor type. In some cases, a sample is somewhat similar to a given tumor subtype, but the score is relatively low, indicating some uncertainty about the validity of the classification. In other cases, there may be comparable scores for more than one tumor type, suggesting that the sample might be intermediate between two diagnoses. The authors present

their methylation-based classification as a single diagnosis. It would be helpful if they could include a classification score indicating the certainty of the classification, and scores for other histologies in cases where there is ambiguity.

3) The high-throughput drug screen is the centerpiece of the study, but the description of the various pilot screens leading up to the final screen is extremely confusing, and the rationale for the drugs, cells and format (single dose vs. dose response) used at each step is difficult to discern. They use a single cell line to compare growth under 2D and 3D conditions, doing dose responses with 53 compounds. Then they use 9 cell lines to test 1134 compounds at a single concentration. They identify 89 top hits from this screen, but instead of just proceeding with those hits, they add 215 additional compounds (for a total of 304), and perform dose responses on 4 cell lines. Identifying 58 potent compounds from this screen, they then add 35 more compounds (for a total of 93) and screen those on all 16 cell lines. This complex series of steps may reflect the process they went through, but it's not clear that all of these steps are necessary or helpful to share with the reader. I would suggest that the authors streamline this section (and the associated data), and just present the essential steps. If they feel it is important to keep everything in, they should clearly name each step of the screen, include a flowchart with those names to help the reader follow the process, and when presenting the data for each step (in Supp Table 5), explain at the top of the table exactly what was done.

4) The results of the final (93-compound) screen are presented in Figure 6a-d, but none of these panels is very easy to interpret. The heatmap in Figure 6a is row-normalized, and this has the effect of highlighting the line with the strongest response to each drug. Unfortunately, one of the lines (SUDIPGXXI) turns out to be the most sensitive to almost all the drugs, and this makes it difficult to glean any useful information about the sensitivity of the other lines (most of the heatmap looks like faint blue and orange lines, without any obvious pattern). The authors should consider an alternative approach to normalization or clustering, e.g. one that highlights the most effective drugs for each cell line, rather than the most sensitive cell lines for each drug. In addition, only a subset of the drugs is labeled on the right side of the heatmap, and it is not clear why these particular drugs are labeled. If they are trying to highlight particular classes of drugs, they should consider color-coding those drug classes. Finally, the graphs in Figures 6b-d have too many lines to be interpretable. The gray line (representing HABS) is often difficult to see; there seems to be a purple line, which is not defined in the legend; and there are several shades of green whose significance is not clear. It would be better to present 4-5 representative curves to convey whatever point they are trying to make.

5) The description of the Figure 6a "Boxes" on pp. 9-10 is also difficult to follow. The discussion of Box 1 starts with the statement that "Tumor cell lines were generally more sensitive to both control cell lines to the compounds in Box 1." In contrast, there is no mention of the significance (i.e. pattern of responsiveness) associated with Boxes 2 and 3. In fact, the drugs in all 3 boxes seem to be highly effective against the lines on the right of the heat map, modestly effective against the lines in the middle, and largely ineffective against the lines on the left (this is, of course, a consequence of how

the clustering was done). If there is no real difference in the pattern of responsiveness, there doesn't seem to be any reason to divide these compounds up into 3 boxes.

6) One major gap in the manuscript is integration of the drug screening results with the methylation, DNA sequencing and RNA sequencing presented in the first part of the paper. Were any of the drug sensitivities predictable from mutations? Were pathways found to be active based on RNA analysis associated with sensitivity to certain classes of drugs? Did the efficacy of epigenetic drugs correlate with particular patterns of DNA methylation? And if there were groups of cell lines that showed similar patterns of drug sensitivity, was there any information the DNA or RNA that could explain these similarities? The fact that "cell lines with shared histone H3 mutation status did not cluster as a separate group" (p. 9) is very interesting, but it would be helpful to know if any other genetic, epigenetic or transcriptional characteristics correlated with drug response. Answering some of these questions would markedly increase the impact of this paper.

7) The synergy testing in Figure 7 could be better explained. In particular, there is no discussion about the significance of the different shapes of the curves (convex for DIPGX37 and DIPGX29, more vertical for DIPGX7 and iASTRO).

8) It is unfortunate that the in vivo survival studies were only done on one cell line (DIPGX37, Figure 9). Given their large bank of cell lines and PDXs, it would have been helpful to see the effects of this drug combination on a few more lines, to get a sense of whether it is effective on most DIPGs and HGGs, or on just a subset. If the authors have already done additional in vivo experiments, it would be great to include them in the manuscript. If this is not possible (knowing how long these studies take, I am not suggesting that it is critical to perform them), the authors should at least discuss this issue based on their in vitro data.

Reviewer #2 (Remarks to the Author):

In this study, He et al describe a collection of new cell, organoid and PDOX models of pediatric high-grade gliomas. The models are characterized using a variety of genomic and histopathology methods and compared to existing cell-based models of the same diseases. The authors also conduct a drug screen to seek out hints for novel therapeutic interventions.

The work provides a very nice collection of new cell models for these diseases. The genomic characterization is well done and the web-based interface is very helpful. In fact, an expanded characterization of these cell lines alone would be a highly valuable contribution to the literature.

The drug screens are fine (with some noted comments/recommendations below). The authors ultimately explore the combinations of MEKi + PI3Ki as a possible, promising drug combinations for further exploration. The in vitro data (synergy as determined by the BRAID model) and in vivo outcomes demonstrating a survival benefit are promising for at least 2 of the cell/PDOX models. As a commentary: the expansion of these models is so critical. The heterogeneity of drug responses shown in this study really underscores that. Our lab see similar outcomes. The authors are really commended for their efforts to expand the # of available models.

The main issue with the work is the lack of mechanistic understanding of the MEKi + PI3Ki synergy. There are multiple reports informing on MEKi + PI3Ki in multiple cancer types. These studies often implicate feedback loops or adaptive signaling amplification in the present of either a MEKi or a PI3Ki alone. Several studies explore the mechanistic implications of RAS/MEK/ERK blockade + PI3K/AKT/mTOR blockage in significant detail with surprising implications for therapy design. And each cancer is, of course, unique. So the lessons from, say, lymphoma are not guaranteed to translate to HGG.

Understanding the MOA rationale is paramount (particularly given the revealed heterogeneity in drug responses). Any possible translation of MEKi + PI3Ki is dependent on a deep understanding of the genomic/mechanistic landscape that primes the cancer cell to respond to dual blockade of these pathways. If the in vitro and in vivo models can yield that insight and, down the road, those lessons can be recognized within in situ disease then the opportunity for translation exists. Without it we are just guessing (even with good drug screening data). The authors are strongly encouraged to dig into the MOA details beyond what they show in figure 8.

Minor comments:

Move the histopathology figure 3 to the SI

Proliferation signatures down in the patient samples. Suggestive that drugs targeting cells in active cycling mode is a dead end? Do the PDOX and cell lines mis-inform us? Does the community pursue cell cycle modulators too frequently in HGGs? Some commentary in the discussion would be welcome.

This sentence on Page 7 could be worded much better: Genes that were upregulated in PDOX models were significantly enriched for hallmark gene sets associated with cell cycle progression (adj $p < 2.2E-16$), while downregulated genes in PDOX compared to patient tumors were enriched for gene sets associated with inflammatory response (adj $p < 2.2E-16$) (Supplementary Table 3).

In vitro assays were conducted over 7 days. The authors should justify this choice. Several drug pharmacologies require longer exposure windows to show a phenotypic response (EZH2i, CDK4/6i). Other agents should elicit a cell response more quickly. While it's not a universal rule: for drugs that target a cell intrinsic target or pathway, the faster a drug can elicit a response the better. A 7-day constant exposure is difficult to replicate for many (most) small molecule drugs. The authors should also provide additional details. Were drugs re-applied at any time? Was additional media added at any time?

Cells plated as adherent cultures versus tumorspheres showed similar response to 53 drugs (supp fig 5). The authors mention that validations showed reproducibility in terms of proliferation kinetics (although that data isn't shown). Does that include a comparison of the growth rate of the adherent-vs-tumorsphere cultures? Furthermore, the drugs in supp fig5 are broken down by class (GPCR/Ion channel, epigenetic, kinase, DNA damage). Were metabolic and cell cycle targeting agents included? Drugs of this nature may show a more divergent dataset.

Ultimately, the authors use adherent cultures. Were key hits vetted using dose-response in the equivalent tumorsphere culture models?

Supplementary figure 7 should be replaced with a table.

Characterizing the response of DIPGX37c ($EC_{50} = 1 \mu M$) as 'strong' is overly subjective.

Reviewer #3 (Remarks to the Author):

This manuscript describes recent efforts to assemble a repository of PDX and cell line models from rare pediatric high-grade gliomas (pHGG), with the ultimate goal of facilitating improved preclinical therapeutic testing. 21 novel pHGG PDX models are reported along with 8 matched cell lines. Methylation profiling and genomic analyses reveal maintenance of core molecular features in these reagents. A subset of the cell lines is then used, along with others, in a drug screen to identify viable candidate compounds for further study. Combined PI3K/MAPK inhibition is then studied in more detail in vitro, and then in vivo, with the latter studies employing two PDX models. Findings from these investigations are available for public access via a customized web portal, and access to cell lines and PDX reagents is offered to the larger scientific community.

This work establishes several disease-relevant pHGG models, whose abundance is generally lacking in the field. The extent to which these reagents will markedly improve efforts to target pHGG, however, is unclear. The drug screening performed by the authors does not appear to have identified particularly promising novel leads. The PI3K and MAPK pathways subjected to the most intensive interrogation have long been implicated in glioma biology, and their targeting thus far has been largely unsuccessful in human trials. Moreover, it is unclear whether the number of PDX models and cell lines available in this set, while impressive, is sufficient to link therapeutic efficacy to specific molecular features. Additional comments are provided below.

- How do the various drugs highlighted by the authors in the results section compare with more conventional approaches (alkylating agents for instance) in their impact on cell line growth and specificity with regard to pHGG? Alkylating agents, including temozolomide, appear to have been included in at least one of the drug screens.
- How stable are these cell lines and PDX reagents? To what extent do they evolve with passaging, or with being grown in adherent monolayers? How do genomic, transcriptional, and methylation profiles change? Such information is crucial to establishing the robustness of these models.

NCOMMS-20-23075

He et al., Patient-Derived Orthotopic Xenografts and Cell Lines from Pediatric High-Grade Glioma Recapitulate the Heterogeneity of Histopathology, Molecular Signatures, and Drug Response

Point-by-Point Response to Reviews

We thank the reviewers for their helpful comments and suggestions that have improved our manuscript, and address each of the points from their reviews directly below.

Reviewer #1:

We thank the reviewer for noting *“The studies described here represent an extraordinary effort that will be of significant benefit to the field of pediatric oncology. The generation of PDOXs and cell lines, the detailed characterization of these lines, and the database that shares these resources with the community, will certainly help move the field as a whole forward.”*

We also appreciate the reviewer’s recommendations to improve the presentation of data and analysis.

1. *The supplementary tables present important information, but in many cases they are hard to follow. Each table should have a legend incorporated at the top, instead of just in a separate Supplementary Information file. The legend should clearly define the abbreviations used in each of the column headings so that the reader can follow what is being presented. Supp Table 5 is particularly challenging; even with the information provided in the Supplementary info file it is hard to know what each of the tabs are, and to interpret the contents of those tabs. As another example, Supp Table 7 has multiple abbreviations that are not defined. Making all of the tables more user-friendly is essential to allow readers to understand and benefit from the authors’ work.*

As recommended, we added legends for each of the supplementary tables. We edited Supplementary Table 5 for clarity by adding legends to every worksheet. However, we kept the four worksheets (Definitions_2Dvs3D comparison, Definitions_pHGG DR screens, Definitions_Combination Raw, Definitions_Combination Fits) to define column headers because we felt that including all column header definitions in the corresponding worksheet legend would be too difficult to follow. In addition, we have edited the column definitions to improve clarity.

2. *The authors present their methylation-based classification as a single diagnosis. It would be helpful if they could include a classification score indicating the certainty of the classification, and scores for other histologies in cases where there is ambiguity.*

We added a new Supplementary Table 1b to include the requested information.

3. *The high-throughput drug screen is the centerpiece of the study, but the description of the various pilot screens leading up to the final screen is extremely confusing, and the rationale for the drugs, cells and format (single dose vs. dose response) used at each step is a difficult to discern...I would suggest that the authors streamline this section (and the associated data), and just present the essential steps. If they feel it is important to keep everything in, they should clearly name each step of the screen, include a flowchart with those names to help the reader follow the process, and when presenting the data for each step (in Supp Table 5), explain at the top of the table exactly what was done.*

We agree that the presentation, which reflected the process over time as we developed additional cell lines and incorporated newly available compounds, was confusing. As suggested, we significantly streamlined the screening narrative to focus on the 93 compounds screened across the 14 pHGG models and 2 control cell lines. We note that “an additional 246 compounds, sampling a broader range of drug mechanisms of action, were screened in DR format in 4 exemplar pHGG models representing different histone sub-types. Results from all stages of these HTS studies are presented in Supplementary Table 5 and are available online for exploration in the Pediatric Brain Tumor Portal (<https://pbtp.stjude.cloud>)” (p. 8-9).

The data from the different generations of screening is valuable, therefore we kept all associated data in Supplementary Table 5 and separated each generation of the screen into different spreadsheets with a clear legend explaining the numbers of compounds and cell lines tested.

Importantly, all of the screening data is also available in the interactive online portal PBTP.stjude.cloud.

4. The results of the final (93-compound) screen are presented in Figure 6a-d, but none of these panels is very easy to interpret. The heatmap in Figure 6a is row-normalized, and this has the effect of highlighting the line with the strongest response to each drug. Unfortunately, one of the lines (SUDIPGXXI) turns out to be the most sensitive to almost all the drugs, and this makes it difficult to glean any useful information about the sensitivity of the other lines (most of the heatmap looks like faint blue and orange lines, without any obvious pattern). The authors should consider an alternative approach to normalization or clustering, e.g. one that highlights the most effective drugs for each cell line, rather than the most sensitive cell lines for each drug...Finally, the graphs in Figures 6b-d have too many lines to be interpretable. It would be better to present 4-5 representative curves to convey whatever point they are trying to make.

We thank the reviewer for these helpful suggestions and completely reformatted the data presentation. We replaced the previous Figure 6 and Supplementary Figure 7 with new figures to emphasize the main findings from the 93 compounds screen in 16 models. Figure 6a now reports the distribution of normalized drug AUC (drug AUC minus the median AUC for that drug across all models) for each model. This plot highlights the most efficacious drugs for each model in terms of selectivity. Moreover, this plot clearly shows that the distribution of responses for SUDIPG-XXI was more sensitive than other lines to nearly every drug tested and supports our rationale to exclude the model in the additional figures so that differences among the other models are more clearly visualized. Figure 6b compares the distribution of AUC values across all 15 models (SUDIPG-XXI was excluded) for “drugs of interest”: compounds identified in Figure 6a and other drugs in the set that have been evaluated in clinical trials or implicated as promising agents in preclinical studies for adult or pediatric glioma. This plot allows the reader to easily compare the average efficacy and range of efficacies across the models for each drug. Select dose responses are presented in Figure 6c. Here, we have used solid and dotted black lines to represent the two control cell lines HABS and iAstro, respectively; colored lines for pHGG models we want to highlight; and gray lines for all other pHGG models. This way of plotting the dose responses makes it easier to understand the point we are trying to make for each drug. Finally, the heatmap in Figure 6d has been simplified by focusing on the 25% most active drugs, reporting z-score as opposed to delta AUC, and color-coding drug classes as recommended by the reviewer. All rows are labelled with drug name, and two clusters enriched for compounds operating by similar mechanisms of action (proteasome inhibitors and epigenetic modulators) are highlighted.

We also note on p. 9 that the interactive features of the data portal PTBP.stjude.cloud allow the user to query by drug class, specific compound name, tumor subgroup, or tumor cell lines, and to visualize results in multiple formats including data on the range of responses to selected compounds, the sensitivity to all tested drugs for selected cells lines, and customized overlay dose-response curves. A

mouse over feature also allows the user to identify outliers and view compound information, dose response values and the associated dose response curve.

5. *The description of the Figure 6a “Boxes” on pp. 9-10 is also difficult to follow.*

We replaced this figure, as recommended in point 4 above. The revised heatmap, in revised Fig. 6d and described on p. 11 clearly delineates two clusters of compounds showing different patterns of selectivity, with each cluster enriched for molecules acting by the same MOA. Box 1 is enriched for proteasome inhibitors and includes some pHGG cell lines with greater resistance compared to normal astrocytes. Box 2 is enriched for compounds targeting transcriptional dependencies and two dual activity compounds including PI3K inhibition. This group of compounds showed a significant lack of efficacy against the iAstro control cell line, therefore showing some selectivity for HGG.

6. *One major gap in the manuscript is integration of the drug screening results with the methylation, DNA sequencing and RNA sequencing presented in the first part of the paper. Were any of the drug sensitivities predictable from mutations? Were pathways found to be active based on RNA analysis associated with sensitivity to certain classes of drugs? Did the efficacy of epigenetic drugs correlate with particular patterns of DNA methylation? And if there were groups of cell lines that showed similar patterns of drug sensitivity, was there any information the DNA or RNA that could explain these similarities? The fact that “cell lines with shared histone H3 mutation status did not cluster as a separate group” (p. 9) is very interesting, but it would be helpful to know if any other genetic, epigenetic or transcriptional characteristics correlated with drug response. Answering some of these questions would markedly increase the impact of this paper.*

We report drug screening results in a sizable collection of 14 pHGG models including eight new cell lines derived by our group (i.e., SJ-DIPG and SI-HGG cell lines), six cell lines from the Monje group that have been used in multiple studies and can act as internal references to compare data from different groups, and two types of normal astrocyte controls. Together, this collection represents significant heterogeneity that poses a challenge for pHGG therapeutics and provides a realistic view of the diversity of drug responses. We agree that the ultimate goal of such studies in addition to identifying agents that show efficacy in some tumors is to identify features that may reliably predict sensitivity and resistance. Unfortunately, consistent predictive patterns were not readily identified. We believe that this is a function of the large collection of cell lines that were used and that it is an important and realistic reflection of the clinical challenge of pHGG.

This is more directly addressed in the revised manuscript. To investigate whether expression signatures for MAPK signaling were predictive of response to pathway inhibition, we evaluated the MAPK pathway activation score (MPAS) which is based on expression of a select set of genes that correlate with sensitivity to MAPK pathway inhibitors in cell lines from some, but not all tumor types (1). We determined the MPAS score in the 14 cell lines evaluated in our drug screen but found no correlation between MPAS score and response to trametinib (Pearson correlation -0.34) (Supplementary Fig. 8b), showing substantial heterogeneity in response without a simple relationship to gene expression signatures. This is included on p. 10 in the revised manuscript. We also used single sample GSEA to evaluate enrichment of GSEA Hallmark Gene Sets in our HGG models, including the PI3K-mTOR and KRAS gene sets, and found no significant associations with response to selective inhibitors within these pathways (not shown). We showed that the anti-metabolite GMX-1778, which disrupts the regeneration of NAD⁺ via NAMPT, demonstrated strong efficacy in all cell lines, but did not show enhanced sensitivity in a line with PPM1D mutation as predicted by a previous report (p. 10). Tumors with different histone H3 mutations are readily classified by their genome-wide DNA methylation

signatures but were not strongly predictive of epigenetic drug efficacy. We showed groups of tumors with similar sensitivity or resistance to particular drugs (new Fig. 6c,d) but could not readily identify a robust genetic or expression signature associated with this. The only two H3 WT pHGG cell lines were both much less sensitive to CDK9 inhibition than almost all H3 mutant tumors (Fig. 6c), but the H3 WT tumors were from different DNA methylation subgroups. This is an intriguing result that should be investigated with additional models as they become available.

Taken together, our cohort is large enough to show that simple correlations are unlikely to be predictive. This is consistent with the clinical challenge of pHGG and highlights the importance of providing these new models with associated molecular characterization and drug response data as a resource for the field to test hypotheses. We address this directly with this addition to the discussion on page 17: “While we did not observe a simple predictive relationship between DNA methylation subgroup and compounds inhibiting a particular MOA, our results showed efficacy of inhibitors of transcription and translation across all pHGG subgroups (Fig. 6b and c). Importantly, we also found that each cell line showed selective responses to specific compounds (Fig. 6a) that were not predicted by simple correlation with DNA methylation subgroups or mutations in signature pHGG genes, indicating that there are clear but heterogeneous vulnerabilities for these tumors. Thus, major advances in the treatment of this heterogeneous group of intractable brain tumors will require significant new insights into mechanisms driving disease pathogenesis and even more extensive preclinical testing to identify more reliable predictors of these selective vulnerabilities. Models with a detailed molecular characterization that can be experimentally manipulated and studied *in vitro* and *in vivo* provide powerful tools needed to address this challenge.”

7. *The synergy testing in Figure 7 could be better explained. In particular, there is no discussion about the significance of the different shapes of the curves (convex for DIPGX37 and DIPGX29, more vertical for DIPGX7 and iASTRO).*

We revised the section in the manuscript on p. 12 describing the combination screen to better explain the significance of the different curve shapes. “When considering the efficacy of the combination (IAE_{50}), synergy made a major contribution in the case of SJ-DIPGX37c and SJ-DIPGX29c, as indicated by the curvature of the 50% (black) and 90% (white) cell viability isoboles. In contrast, the IAE_{50} value for SJ-DIPGX7c was driven solely by paxalisib: the 50% and 90% isoboles run parallel to the y-axis because mirdametinib exerts little cytotoxicity on its own and fails to potentiate the activity of paxalisib. While the synergy is highest in iAstro ($\kappa = 8.8$), as evidenced by the clear shift in the 50% isobole with increasing mirdametinib concentrations, both drugs are weakly cytotoxic on their own and their interaction is insufficient to induce high combined efficacy ($IAE_{50} = 1.0$).”

8. *It is unfortunate that the in vivo survival studies were only done on one cell line (DIPGX37, Figure 9). Given their large bank of cell lines and PDXs, it would have been helpful to see the effects of this drug combination on a few more lines, to get a sense of whether it is effective on most DIPGs and HGGs, or on just a subset. If the authors have already done additional in vivo experiments, it would be great to include them in the manuscript. If this is not possible (knowing how long these studies take, I am not suggesting that it is critical to perform them), the authors should at least discuss this issue based on their in vitro data.*

We appreciate the reviewer’s acknowledgment of the long timeline required for *in vivo* experiments, which has been made more difficult with additional COVID-associated restrictions. We did show *in vivo* pharmacodynamic responses in two PDOX lines matched to *in vitro* studies to show that MEK inhibition

by mirdametinib was consistently greater and showed greater synergy with paxalisib in SJ-DIPGX37 compared to SJ-DIPGX7 *in vitro* and *in vivo*. As suggested, we added to the discussion on p16.

Reviewer #2:

We thank the reviewer for noting the “*very nice collection of new cell models for these diseases. The genomic characterization is well done and the web-based interface is very helpful*”, and “*the expansion of these models is so critical. The heterogeneity of drug responses shown in this study really underscores that. Our lab see similar outcomes. The authors are really commended for their efforts to expand the # of available models.*”

We address the reviewer’s specific questions and comments here:

1. *The main issue with the work is the lack of mechanistic understanding of the MEKi + PI3Ki synergy... Understanding the MOA rationale is paramount (particularly given the revealed heterogeneity in drug responses). Any possible translation of MEKi + PI3Ki is dependent on a deep understanding of the genomic/mechanistic landscape that primes the cancer cell to respond to dual blockade of these pathways. If the in vitro and in vivo models can yield that insight and, down the road, those lessons can be recognized within in situ disease then the opportunity for translation exists. Without it we are just guessing (even with good drug screening data). The authors are strongly encouraged to dig into the MOA details beyond what they show in figure 8.*

We agree that a goal for all rational therapeutic design is to fully understand the mechanisms of action for accurate prediction of the ideal drug combination for each tumor. Our results highlight the significant heterogeneity between models, consistent with the complex heterogeneity found in pHGG patients. As mentioned in the response #6 to Reviewer 1, we evaluated the MAPK pathway activation score (MPAS) (revised Supplementary Fig. 8b) as well as enrichment of Hallmark GSEA gene sets for PI3K/mTOR and KRAS pathways but found no correlation with sensitivity to selective inhibitors of these pathways. We also investigated the kinetics of transient inhibition of RAS-ERK signaling following PI3K inhibition, which Will et al (2) reported was an important interplay between pathways that influenced efficacy. However, this did not correlate with sensitivity between models (therefore not included in the revised manuscript). We appreciate the editor’s recognition that a clear mechanism of action for these drugs, tested here as proof-of-principle to evaluate whether in vitro results were reflected in pharmacodynamic responses in vivo for drugs where target engagement could be assessed, is beyond the scope of this report of molecular and chemical sensitivity characterization of a significant group of new pHGG models.

MINOR POINTS:

2. *Move the histopathology figure 3 to the SI*

We believe that including histopathological assessment of the PDOX models is an important component of our characterization to compare the fidelity of the models to the original patient tumor. We are happy to move this to the supplementary data section if the editor also prefers this but given the appreciation of other reviewers for the characterization of our models, we have currently kept this representative example of histopathology comparison as Fig. 3.

3. *Proliferation signatures down in the patient samples. Suggestive that drugs targeting cells in active*

cycling mode is a dead end? Do the PDOX and cell lines mis-inform us? Does the community pursue cell cycle modulators too frequently in HGGs? Some commentary in the discussion would be welcome.

Figure 5 compares gene expression signatures of PDOX models, cell lines, and patient tumors from which the models were derived. Because the heatmap is Z-scored, it emphasizes the differences among these samples, which we wanted to include to demonstrate the heterogeneity represented by our models within pHGG. If normal brain expression profiles had been available, we expect that all tumor samples would show clear proliferative signatures. This is supported by the fact that the patient tumors from which the PDOX models were established were classified by histopathology as high-grade glioma, which is characterized by brisk mitotic index. So, the patient tumors still contain an appreciable proportion of proliferative cells. Brabetz et al also noted higher proliferative expression signatures in PDOX models compared with patient tumors (cited on p. 15). We added to the discussion on p. 15, “Expression differences between the patient tumors and derived PDOX may be due to greater tumor purity in the PDOX expression profiles after removing mouse reads from non-tumor cells within the sample. However, the process of establishing cell lines or PDOXs may select for more proliferative populations. While such selection could lead to an overestimate of the effects of cell cycle inhibition in pHGG models, such bias was not observed in the control cell lines that proliferated more rapidly than the tumor cells.”

4. sentence on Page 7 could be worded much better: Genes that were upregulated in PDOX models were significantly enriched for hallmark gene sets associated with cell cycle progression (adj $p < 2.2E-16$), while downregulated genes in PDOX compared to patient tumors were enriched for gene sets associated with inflammatory response (adj $p < 2.2E-16$) (Supplementary Table 3).

We agree and re-worded the statement: “analysis of genes that were differentially expressed between PDOX and patient samples across the entire matched cohort ($|\log FC| > 1$, adj. $p < 0.05$) showed upregulation of genes associated with cell cycle progression (adj. $p < 2.2E-16$) and downregulation of genes associated with inflammatory response (adj. $p < 2.2E-16$) in PDOX (Supplementary Table 3).” (p.7)

5. In vitro assays were conducted over 7 days. The authors should justify this choice. Several drug pharmacologies require longer exposure windows to show a phenotypic response (EZH2i, CDK4/6i). Other agents should elicit a cell response more quickly. While its not a universal rule: for drugs that target a cell intrinsic target or pathway, the faster a drug can elicit a response the better. A 7-day constant exposure is difficult to replicate for many (most) small molecule drugs. The authors should also provide additional details. Were drugs re-applied at any time? Was additional media added at any time?

Choosing a 7-day exposure balances a design to allow cells to cycle at least once during the time they are exposed to drug versus the need to re-drug. It is possible that some drugs might have degraded during the experiment; however, we chose not to re-drug because this action would introduce logistical hurdles and additional variability. We did not observe plate artifacts that would signal evaporation of media, and therefore, we did not add media during the drug screening experiments. We clarified this procedure on p. 24 in the methods section by adding “Media and drug was not changed during the 7-day incubation to avoid logistical difficulties including cell loss with media change.”

Some of the strongest hits identified in our screens were epigenetic regulators including HDAC inhibitors, but we agree that some epigenetic modifiers may require longer drug exposure times to detect significant effects. This would require both media change and re-application of drugs that was not feasible in the high-throughput format that we used and would be an important follow-up study to

focus on particular pathways or inhibitors. We added this to the discussion on p. 15: “The 7-day assay performed in our study could underestimate the efficacy of some epigenetic modulatory compounds, which may take longer to manifest than compounds inducing acute cytostatic or cytotoxic responses in growth inhibitory assays.”

6. *The authors mention that validations showed reproducibility in terms of proliferation kinetics (although that data isn't shown). Does that include a comparison of the growth rate of the adherent-vs-tumorsphere cultures? Furthermore, the drugs in supp fig5 are broken down by class (GPCR/Ion channel, epigenetic, kinase, DNA damage). Were metabolic and cell cycle targeting agents included? Drugs of this nature may show a more divergent dataset.*

Overall, in our initial pilot work to establish our screening approach, we decided to screen the cells as adherent cultures after we found that gene expression signatures of adherent cultures were as representative of *in vivo* tumor growth as the tumorsphere cultures and showed strong correlation with drug response as shown in Supplementary Figure 6. We then used growth curves to establish optimal seeding numbers for 384-well plates for each of the cell lines using adherent conditions. In general, we do not see significant differences in cell numbers between large-scale cultures grown as tumorspheres compared with adherent cultures, but we did not test this in a systematic way for each line in the 384 well format used for screening.

We did screen several metabolic and cell cycle targeting agents. In the screen testing 93 compounds across 16 models, we profiled the following: (a) metabolic inhibitors GMX-1778 (NAMPT), everolimus (mTOR), and omipalisib (PI3K/mTOR); and (b) cell cycle inhibitors abemaciclib (CDK4/6) and MK-1775 (WEE1). A wider array of metabolic and cell cycle targeting agents were included in the screen of 246 tested against 4 pHGG models. Metabolic inhibitors included: (a) anti-metabolites: leflunomide (DHODH), AVN-944 (IMPDH2), DFMO (ODC), inhibitors of purine biosynthesis (azaguanine-8, brivudine, cladribine, clofarabine, fludarabine, mercaptopurine, pentostatin, thioguanine, mycophenolate, cytarabine, gemcitabine); (b) the anti-folates methotrexate, pemetrexed, pralatrexate, pyrimethamine, raltitrexed, trimetrexate, 5-fluorouracil, carmofur, floxuridine; (c) the AKT inhibitor ipatasertib, the mTOR inhibitor sirolimus, and the PI3K inhibitors buparlisib and taselisib; (d) the statin drugs atorvastatin, fluvastatin, lovastatin, mevastatin, pitavastatin, rosuvastatin, simvastatin; (d) and the PPAR, gamma inhibitor pioglitazone. Cell cycle inhibitors included: (a) the cdc42 cell cycle inhibitor ML141 and the CDK4/6 inhibitor ribociclib. Results from both screens are reported in Supplementary Table 5.

Ultimately, the authors use adherent cultures. Were key hits vetted using dose-response in the equivalent tumorsphere culture models?

Supplementary Figure 6 shows very strong correlation when comparing the AUC for 53 compounds tested in dose-response on SJ-DIPGX7c cells grown as adherent cultures or as neurospheres (Pearson correlation 0.994). The compounds tested for this comparison are listed in Supplementary Table 5b and include compounds from multiple classes highlighted in Figure 6 as key hits including HDAC inhibitor panobinostat, kinase inhibitors trametinib, omipalisib, anti-metabolite NAMPT inhibitor GMX-1778, proteasome inhibitor bortezomib, and Hsp90 inhibitor ganetespib among others.

Characterizing the response of DIPGX37c (EC50 = 1 uM) as 'strong' is overly subjective. We edited "strong" to "stronger".

Reviewer #3 (Remarks to the Author):

1. This work establishes several disease-relevant pHGG models, whose abundance is generally lacking in the field. The extent to which these reagents will markedly improve efforts to target pHGG, however, is unclear. The drug screening performed by the authors does not appear to have identified particularly promising novel leads. The PI3K and MAPK pathways subjected to the most intensive interrogation have long been implicated in glioma biology, and their targeting thus far has been largely unsuccessful in human trials. Moreover, it is unclear whether the number of PDX models and cell lines available in this set, while impressive, is sufficient to link therapeutic efficacy to specific molecular features.

We addressed the urgent need for additional *in vivo* models on p. 14: “pHGG remains largely incurable despite decades of clinical trials. Worldwide efforts in studying the molecular basis of this group of tumors has revolutionized our understanding and revealed subgroups based on spatiotemporal tumor occurrence and striking molecular heterogeneity that has been further subdivided by DNA methylation signatures and inter- and intra-tumoral heterogeneity of mutated genes. Relevant *in vitro* and *in vivo* disease models that are founded in the correct developmental origins, recapitulate genetic and epigenetic signatures and represent the significant heterogeneity of pHGG are essential to further our understanding of mechanisms driving tumorigenesis and to identify therapeutic vulnerabilities. Although a growing number of DIPG cell lines have been established and characterized, relatively few of these efficiently engraft in the brain as xenografts for *in vivo* modeling, and there are a much smaller number of cell lines from pediatric gliomas arising outside the brainstem. PDOXs allow researchers to address critical dimensions of tumor biology, including angiogenesis, tumor invasion, and interactions with the tumor microenvironment, including the contribution of nervous system activity, that may strongly influence tumor growth and selective pressures. A recent study to establish a biobank of pediatric brain tumors reported the establishment of 8 new pHGG PDOX models. The 21 new pHGG PDOX models and eight new cell lines reported here are a significant advance and include several rare tumor subtypes with limited available models...”

We address the comment about whether any of the hits are promising on p. 17: “While we did not observe a simple predictive relationship between DNA methylation subgroup and compounds inhibiting a particular MOA, our results showed efficacy of inhibitors of transcription and translation across all pHGG subgroups (Fig. 6b and c). Importantly, we also found that each cell line showed selective responses to specific compounds (Fig. 6a) that were not predicted by simple correlation with DNA methylation subgroups or mutations in signature pHGG genes, indicating that there are clear but heterogeneous vulnerabilities for these tumors. Thus, major advances in the treatment of this heterogeneous group of intractable brain tumors will require significant new insights into mechanisms driving disease pathogenesis and even more extensive preclinical testing to identify more reliable predictors of these selective vulnerabilities. Models with a detailed molecular characterization that can be experimentally manipulated and studied *in vitro* and *in vivo* provide powerful tools needed to address this challenge.”

We wanted to test whether *in vitro* HTS results predicted *in vivo* response. “For these studies, we chose PI3K/mTOR and MEK pathway inhibitors because responses in different pHGG lines varied and target engagement can be reliably detected in tumor tissue.”(p. 11). Such biomarkers are much less well-understood for many of the compounds that were hits in our screen, complicating interpretation of

whether differences between in vivo and in vitro efficacy are due to drug penetrance rather than intrinsic sensitivity in different experimental conditions.

2. *How do the various drugs highlighted by the authors in the results section compare with more conventional approaches (alkylating agents for instance) in their impact on cell line growth and specificity with regard to pHGG? Alkylating agents, including temozolomide, appear to have been included in at least one of the drug screens.*

We appreciate the reviewer's suggestion to comment on the efficacy of temozolomide, which has been used as the backbone of chemotherapy following radiation and surgery for many pHGG patients because of its positive effects in adult glioblastoma. Consistent with the lack of clear clinical evidence that temozolomide (TMZ) provides any survival advantage for pHGG patients, we found a minimal effect of TMZ. We screened 5 drugs that are known to alkylate DNA (chlorambucil, dacarbazine, nimustine, streptozosin, and temozolomide) but found little activity in this set. These results are reported in the worksheet entitled "DR fits (246 cmpds, 4 pHGG)" in Supplementary Table 5. We included temozolomide to the new Fig. 6b to show the striking contrast between TMZ and many of the drugs tested and highlighted as top hits in our screen. We edited the manuscript to include this important comparison: "Likewise, the alkylating agent temozolomide (TMZ), a standard of care in adult gliomas, was ineffective in our tumor models, consistent with the lack of clinical response to TMZ in pHGG. We also tested two other alkylating agents, streptozocin and nimustine, and both were ineffective in our pHGG models (Supplementary Table 5 c and e)." (p. 10-11).

3. *How stable are these cell lines and PDX reagents? To what extent do they evolve with passaging, or with being grown in adherent monolayers? How do genomic, transcriptional, and methylation profiles change? Such information is crucial to establishing the robustness of these models.*

DNA methylation profiles are increasingly incorporated along with histopathology and selective mutation analyses, into the classification of brain tumors. Overall, DNA methylation signatures of PDOX models and cell lines not only preserve the classification designated by DNA methylation signatures (Supplementary Table 1b), but cluster closely together in tSNE plots for genome-wide methylation signatures (Figure 2). As shown in the oncoprint in Fig. 4, most mutations in signature genes associated with pHGG are conserved between the patient tumor, PDOX and cell line. The oncoprint includes PDOX from passage 7 and passage 10 for SJ-DIPGX7 and PDOX from passages 3 and 4 for SJ-DIPGX29. Genome, transcriptome and methylation profiles of the cell lines were performed from nucleic acids extracted from adherent cultures, which are grown in neural stem cell media and on plates coated with extracellular matrix previously shown to preserve the fidelity of glioma stem-like cells, similar to tumorsphere cultures (3) (4, 5). As shown in the new Supplementary Fig. 4, and reported on p. 7, Representative PDOX models retained fidelity of transcriptome signatures over multiple passages (Pearson correlation 0.98, $p < 2.2 \times 10^{-16}$). Cell lines, which represent extensive passaging in neural stem cell growth media, also showed strong fidelity with the matched PDOX models from which they were derived (Pearson correlation from 0.87-0.95, $p < 2.2 \times 10^{-16}$) (Supplementary Fig. 4).

References:

1. Wagle MC, Kirouac D, Klijn C, Liu B, Mahajan S, Junttila M, Moffat J, Merchant M, Huw L, Wongchenko M, Okrah K, Srinivasan S, Mounir Z, Sumiyoshi T, Haverty PM, Yauch RL, Yan Y, Kabbarah O, Hampton G, Amler L, Ramanujan S, Lackner MR, Huang SA. A transcriptional MAPK Pathway Activity Score (MPAS) is a clinically relevant biomarker in multiple cancer types. *NPJ Precis Oncol.* 2018;2(1):7. PMID: PMC5871852.
2. Will M, Qin ACR, Toy W, Yao Z, Rodrik-Outmezguine V, Schneider C, Huang X, Monian P, Jiang X, de Stanchina E, Baselga J, Liu N, Chandarlapaty S, Rosen N. Rapid induction of apoptosis by PI3K inhibitors is dependent upon their transient inhibition of RAS-ERK signaling. *Cancer discovery.* 2014;4(3):334-47. PMID: PMC4049524.
3. Pollard SM, Yoshikawa K, Clarke ID, Danovi D, Stricker S, Russell R, Bayani J, Head R, Lee M, Bernstein M, Squire JA, Smith A, Dirks P. Glioma stem cell lines expanded in adherent culture have tumor-specific phenotypes and are suitable for chemical and genetic screens. *Cell Stem Cell.* 2009;4(6):568-80.
4. Dolma S, Selvadurai HJ, Lan X, Lee L, Kushida M, Voisin V, Whetstone H, So M, Aviv T, Park N, Zhu X, Xu C, Head R, Rowland KJ, Bernstein M, Clarke ID, Bader G, Harrington L, Brumell JH, Tyers M, Dirks PB. Inhibition of Dopamine Receptor D4 Impedes Autophagic Flux, Proliferation, and Survival of Glioblastoma Stem Cells. *Cancer cell.* 2016;29(6):859-73. PMID: PMC5968455.
5. Jiang Y, Marinescu VD, Xie Y, Jarvius M, Maturi NP, Haglund C, Olofsson S, Lindberg N, Olofsson T, Leijonmarck C, Hesselager G, Alafuzoff I, Fryknäs M, Larsson R, Nelander S, Uhrbom L. Glioblastoma Cell Malignancy and Drug Sensitivity Are Affected by the Cell of Origin. *Cell reports.* 2017;18(4):977-90.

REVIEWERS' COMMENTS

Reviewer #2 (Remarks to the Author):

The authors have done an excellent job addressing my previous comments. The only minor issue is that Supplementary Table 5 is not labeled as such; a heading should be added to the excel file for that table. Beyond that, the paper reads beautifully and represents an important and timely contribution to the field.

Reviewer #3 (Remarks to the Author):

In their response, Baker and coworkers address many of the reviewers concerns. Furthermore, the need for a broader mechanistic understanding of the MEKi + PI3Ki combination is agreed to be outside the scope of this manuscript.

As such the authors have addressed the key concerns.

Craig Thomas

Reviewer #4 (Remarks to the Author):

I have no further issues with this manuscript.

We thank all reviewers for their helpful comments.
There was only one minor request for revision of our last submission.

Reviewer #2 (Remarks to the Author):

*The authors have done an excellent job addressing my previous comments. **The only minor issue is that Supplementary Table 5 is not labeled as such; a heading should be added to the excel file for that table.** Beyond that, the paper reads beautifully and represents an important and timely contribution to the field.*

Supplementary Table 5 was re-labeled Supplementary Data 4, according to formatting requests from the editor. We added the heading Supplementary Data 4 to the Excel file.

Reviewers 3 and 4 did not request any revisions.